# Convergence Rate of the Last Iterate of Stochastic Proximal Algorithms

**Kevin Kurian Thomas Vaidyan** [1]   **Michael P. Friedlander** [* 1]   **Ahmet Alacaoglu** [* 1]

## Abstract

We analyze two classical algorithms for solving additively composite convex optimization problems where the objective is the sum of a smooth term and a nonsmooth regularizer. The first algorithm is the proximal stochastic gradient method for a single regularizer; the second is the randomized incremental proximal method, which uses the proximal operator of a randomly selected function when the regularizer is given as the sum of many nonsmooth functions. We focus on relaxing the bounded variance assumption that is common, yet stringent, for getting last iterate convergence rates. We prove the $\widetilde{O}(1/\sqrt{T})$ rate of convergence for the last iterate of both algorithms under componentwise convexity and smoothness, which is optimal up to log terms. Our results apply directly to graph-guided regularizers that arise in multi-task and federated learning, where the regularizer decomposes as a sum over edges of a collaboration graph.

## 1. Introduction

Composite optimization problems with structured regularizers arise throughout machine learning. In multi-task learning, for example, related tasks share statistical structure that can be harnessed through *graph-guided regularizers* (Kim & Xing, 2009): a collaboration graph connects tasks with shared characteristics, and the regularizer encourages their parameters to agree. This formulation similarly manifests in federated optimization, where agents learn related models without centralizing data, and in sensor networks that aggregate spatially correlated measurements. (Smith et al., 2017).

A canonical instance is the *network Lasso* (Hallac et al.,

*Joint last authors. [1]University of British Columbia, Vancouver. Correspondence to: K. K. T. Vaidyan <kevinktv@cs.ubc.ca>, M. P. Friedlander <michael.friedlander@ubc.ca>, A. Alacaoglu <ahmet.alacaoglu@ubc.ca>.

*Proceedings of the 43rd International Conference on Machine Learning*, Seoul, South Korea. PMLR 306, 2026. Copyright 2026 by the author(s).

2015), where each node $i$ maintains parameters $\mathbf{x}_i$ and the regularizer penalizes disagreement between neighbors:

$$g(\mathbf{x}) = \sum_{(i,j)\in\mathcal{E}} w_{ij}\|\mathbf{x}_i - \mathbf{x}_j\|_p.$$

The regularizer naturally decomposes as a sum over edges, $g = \sum_i g_i$, where each $g_i$ involves only the variables at connected nodes.

More generally, consider the regularized optimization problem

$$\min_{\mathbf{x}\in\mathbb{R}^n} \ h(\mathbf{x}) := f(\mathbf{x}) + g(\mathbf{x}), \tag{1.1}$$

where $f\colon \mathbb{R}^n \to \mathbb{R}$ is convex and differentiable, and $g\colon \mathbb{R}^n \to \mathbb{R} \cup \{+\infty\}$ is convex. We assume access to a stochastic gradient oracle that returns unbiased estimates $\nabla f_i$ that satisfy

$$\mathbb{E}[\nabla f_i(\mathbf{x})] = \nabla f(\mathbf{x}) \quad \text{for all } \mathbf{x}.$$

A standard assumption for analyzing stochastic gradient descent (SGD) requires the gradient estimates to have a uniformly bounded variance. This assumption is overly restrictive and fails even for simple problems; see Equation (1.3) for an example.

Garrigos et al. (2025) and Attia et al. (2025) showed that the bounded variance assumption can be avoided for last-iterate analyses of SGD in unconstrained optimization by instead requiring that

$$\text{each } f_i \text{ is convex and } L\text{-smooth.} \tag{1.2}$$

Under this condition, the last iterate of SGD achieves the optimal (up-to-log) $\widetilde{O}(1/\sqrt{T})$ rate. The extension to *regularized* problems, however, does not follow directly: the proximal operator is nonlinear, so the iterates no longer admit the closed-form expansion that these analyses utilize.

The workhorse method for regularized problems in machine learning is the proximal SGD (Duchi & Singer, 2009), which iterates as

$$\mathbf{x}_{t+1} = \operatorname{prox}_{\tau g}(\mathbf{x}_t - \tau\nabla f_{i_t}(\mathbf{x}_t)), \tag{SPGD}$$

where $\operatorname{prox}_g(\mathbf{x}) := \arg\min_{\mathbf{z}}\{g(\mathbf{z}) + \frac{1}{2}\|\mathbf{x} - \mathbf{z}\|^2\}$ and $i_t$ is selected i.i.d. at every iteration with respect to a fixed

distribution $D$. (More generally, the step size $\tau$ may vary with each iteration, but our analysis focuses on constant step sizes.) Our purpose is to extend the last-iterate guarantees of Attia et al. (2025); Garrigos et al. (2025) to this setting.

**A classical example.** Let us start with perhaps the most standard, textbook example of a regularized problem, Lasso, or linear least squares regression with $\ell_1$ regularization, given as

$$\min_{\mathbf{x}\in\mathbb{R}^n} \frac{1}{2N}\sum_{i=1}^{N}(\langle\mathbf{a}_i,\mathbf{x}\rangle - b_i)^2 + \lambda\|\mathbf{x}\|_1,$$

where $\mathbf{a}_i \in \mathbb{R}^n$ and $b_i \in \mathbb{R}$. Of course, there are many methods developed for this problem, including variance reduced algorithms (Gower et al., 2020). However, when the number of data points $N$ is extremely large, SGD-based methods are the main choices since they have rates of convergence not depending on $N$, allowing arbitrarily large $N$.

Here, the typical choice of a stochastic gradient is

$$\nabla f_i(\mathbf{x}) = \mathbf{a}_i(\langle\mathbf{a}_i,\mathbf{x}\rangle - b_i). \tag{1.3}$$

Even for this problem, classical assumptions such as the bounded variance (or other distributional assumptions) do not necessarily hold since the quantity $\nabla f_i(\mathbf{x}) - \nabla f(\mathbf{x})$ is linear in $\mathbf{x}$. Hence, the existing results showing the last-iterate convergence rate of proximal SGD from Liu & Zhou (2024) do not apply. Moreover, due to the existence of the $\ell_1$ regularizer, existing results for unconstrained SGD without bounded variance from Attia et al. (2025); Garrigos et al. (2025) also do not apply.

That is, the literature is currently missing the theoretical guarantees for the most standard setup for solving this classical problem in the large-scale regime: proximal SGD with last-iterate as the output. Our results address this gap.

Our main contributions are as follows:

- We prove that the last iterate of *proximal SGD* converges at the optimal (up-to-log) rate $\widetilde{O}(1/\sqrt{T})$ under componentwise convexity and smoothness of $f_i$, with second moment of stochastic gradients bounded only at the solution (Theorem 3.1). As a special case, we obtain the same rate for projected SGD.

- When the regularizer decomposes additively as $g(\mathbf{x}) = \sum_{i=1}^{m} g_i(\mathbf{x})$, the proximal map $\mathrm{prox}_g$ may be intractable even when each $\mathrm{prox}_{g_i}$ admits closed form. We prove the $\widetilde{O}(1/\sqrt{T})$ last-iterate rate for the *randomized incremental proximal method* (Bertsekas, 2011)

$$\mathbf{x}_{t+1} = \mathrm{prox}_{\tau m g_{j_t}}(\mathbf{x}_t - \tau\nabla f_{i_t}(\mathbf{x}_t)), \quad \text{(RIPM)}$$

where $j_t$ is selected uniformly at random and $i_t$ is selected as (SPGD) (Theorem 4.1). This implies optimal (up-to-log) rates for stochastic proximal point methods.

- For graph-guided regularizers, such as network Lasso and multi-task learning with task similarity constraints, our results lead to last-iterate convergence for the BlockProx algorithm (Lin et al., 2025). We thus establish optimal rates where only ergodic convergence was previously known without bounded variance.

- We provide numerical experiments confirming that the last iterate outperforms averaged iterates in practice for proximal SGD.

**A note on presentation.** Although (RIPM) generalizes (SPGD), we analyze the simpler algorithm first in Section 3 and treat the incremental setting in Section 4. This ordering serves three purposes: *(i)* Handling randomly selected proximal operators in (RIPM) alongside stochastic gradients requires additional analytical machinery, so we focus first on the case where only gradients are stochastic (the algorithm (SPGD)) so that we can delineate the particular difficulties in the analysis and our solutions. *(ii)* Projected and proximal SGD algorithms are interesting in their own right as classical tools for solving important classes of regularized and constrained optimization problems. *(iii)* The generality of (RIPM) requires further assumptions for the problem template in Section 4 and hence the results are not directly comparable.

## 2. Problem setting and main assumptions

The standard bounded variance assumption in stochastic optimization requires the following inequality to hold uniformly over all $\mathbf{x}$: $\mathbb{E}[\|\nabla f_i(\mathbf{x}) - \nabla f(\mathbf{x})\|^2] \leq \sigma^2$. This overly restrictive condition fails even for unconstrained linear least-squares, see for example (1.3). For unconstrained smooth problems (i.e., $g = 0$), recent work has shown that it suffices to assume finite second moments at the solution:

$$\mathbb{E}\|\nabla f_i(\mathbf{x}^*)\|^2 = \sigma_*^2 < \infty \quad \text{for some } \mathbf{x}^* \in \arg\min f. \tag{2.1}$$

When multiple minimizers exist, this bound holds for every minimizer (Garrigos & Gower, 2024, Lemma 8.23). For the composite problem (1.1) with $g \neq 0$, we impose the analogous condition at minimizers of $h = f + g$. The analysis, however, requires new tools because the proximal operator introduces additional structure not present in the unconstrained setting.

We consider two settings for problem (1.1):

1. *Proximal SGD.* The smooth component $f$ is an expectation over a distribution $D$ from which we can draw i.i.d. samples:

$$f(\mathbf{x}) = \mathbb{E}_{i\sim D}[f_i(\mathbf{x})], \tag{2.2}$$

and we have access to the full proximal operator $\mathrm{prox}_{\tau g}$.

2. *Incremental proximal.* In addition, the regularizer decomposes as $g(\mathbf{x}) = \sum_{j=1}^m g_j(\mathbf{x})$, and we access only proximal operators of individual components $g_j$.

**Assumption 2.1** (Standing assumptions).

1. The solution set $X^* = \arg\min_{\mathbf{x}} h(\mathbf{x})$ is nonempty. We write $\mathbf{x}^*$ for an arbitrary element of $X^*$, $h^* = h(\mathbf{x}^*)$ for the optimal value, and

$$D_*^2 = \min_{\mathbf{x}^* \in X^*} \|\mathbf{x}^* - \mathbf{x}_0\|^2,$$

for the squared distance from the initial point to the solution set. Let also $\mathbb{E}\|\nabla f_i(\mathbf{x}^*)\|^2 = \sigma_*^2 < \infty$ for some $\mathbf{x}^* \in \arg\min h$.

2. Every component function $f_i \colon \mathbb{R}^n \to \mathbb{R}$ is convex and differentiable with $L$-Lipschitz gradient.

3. The regularizer $g \colon \mathbb{R}^n \to \mathbb{R} \cup \{+\infty\}$ is proper, convex, and lower semicontinuous.

For the incremental proximal setting, we impose additional structure on the regularizer.

**Assumption 2.2** (Decomposable regularizer).

1. Each $g_j \colon \mathbb{R}^n \to \mathbb{R}$ is proper, convex, and lower semicontinuous.

2. Each $g_j$ is $L_g$-Lipschitz: $|g_j(\mathbf{x}) - g_j(\mathbf{y})| \le L_g \|\mathbf{x} - \mathbf{y}\|$ for all $\mathbf{x}, \mathbf{y} \in \mathbb{R}^n$.

The Lipschitz condition on $g_j$ controls the variance introduced by proximal stochasticity. This requirement is common in methods that access nonsmooth terms randomly, in the non-strongly convex case, see for example (Asi & Duchi, 2019; Bertsekas, 2011; Cai & Diakonikolas, 2025). We will share more insights regarding the need for this condition in Section 4.2.

**Notation.** Given iterates $(\mathbf{x}_t)$, we denote as $\mathcal{F}(\mathbf{x}_0, \dots \mathbf{x}_t)$ the $\sigma$-algebra generated by $\mathbf{x}_0, \dots \mathbf{x}_t$. The standard notation $\mathbb{E}_t[\cdot]$ is used for the expectation conditioned on $\mathcal{F}(\mathbf{x}_0, \dots \mathbf{x}_t)$, that is, $\mathbb{E}[\cdot|\mathcal{F}_t] = \mathbb{E}_t[\cdot]$.

### 2.1. Variance control via co-coercivity

This section presents the mechanism that relaxes the bounded variance assumption for unconstrained SGD, and explains why the proximal setting requires additional analytical tools. The mechanism relies on co-coercivity: convexity and $L$-smoothness of each $f_i$ imply (Nesterov, 2018, Thm. 2.1.5)

$$\|\nabla f_i(\mathbf{x}) - \nabla f_i(\mathbf{y})\|^2 \le 2L\big(f_i(\mathbf{x}) - f_i(\mathbf{y}) - \langle \nabla f_i(\mathbf{y}), \mathbf{x} - \mathbf{y} \rangle\big). \tag{2.3}$$

The right-hand side involves only function values, and thus bounds the stochastic gradient norm $\|\nabla f_i(\mathbf{x}_t)\|^2$ without additional assumptions.

Consider standard SGD for unconstrained minimization, where $g \equiv 0$ and let $\mathbf{x}^* \in \arg\min f$:

$$\mathbf{x}_{t+1} = \mathbf{x}_t - \tau \nabla f_{i_t}(\mathbf{x}_t),$$

where $i_t \sim D$ is selected i.i.d., see (2.2).

Expanding the squared distance to a reference point $\mathbf{z}$ gives

$$\|\mathbf{x}_{t+1} - \mathbf{z}\|^2 = \|\mathbf{x}_t - \mathbf{z}\|^2 - 2\tau\langle \nabla f_{i_t}(\mathbf{x}_t), \mathbf{x}_t - \mathbf{z}\rangle + \tau^2 \|\nabla f_{i_t}(\mathbf{x}_t)\|^2. \tag{2.4}$$

Taking expectation over the choice of $i_t$ and applying convexity of each $f_i$ to the inner product yields

$$\mathbb{E}\|\mathbf{x}_{t+1} - \mathbf{z}\|^2 \le \mathbb{E}\|\mathbf{x}_t - \mathbf{z}\|^2 \underbrace{-2\tau(f(\mathbf{x}_t) - f(\mathbf{z}))}_{\text{descent term}} + \underbrace{\tau^2 \mathbb{E}\|\nabla f_i(\mathbf{x}_t)\|^2}_{\text{variance term}}. \tag{2.5}$$

The descent term is negative whenever $f(\mathbf{x}_t) > f(\mathbf{z})$ and thus drives convergence. The variance term, however, is positive, and without an *a priori* bound, it prevents meaningful control of the distance to $\mathbf{z}$.

We now apply the co-coercivity inequality (2.3) with $i = i_t$, $\mathbf{x} = \mathbf{x}_t$, $\mathbf{y} = \mathbf{x}^*$, and use Young's inequality to deduce

$$\|\nabla f_{i_t}(\mathbf{x}_t)\|^2 \le 2\|\nabla f_{i_t}(\mathbf{x}^*)\|^2 + 4L\big(f_{i_t}(\mathbf{x}_t) - f_{i_t}(\mathbf{x}^*) - \langle \nabla f_{i_t}(\mathbf{x}^*), \mathbf{x}_t - \mathbf{x}^*\rangle\big). \tag{2.6}$$

Taking expectation over $i_t$, the inner product vanishes because $\mathbb{E}_{i_t}[\nabla f_{i_t}(\mathbf{x}^*)] = \nabla f(\mathbf{x}^*) = 0$. Multiplying (2.6) by $\tau^2$ and using (2.1) yields the variance bound

$$\tau^2 \mathbb{E}_t\|\nabla f_{i_t}(\mathbf{x}_t)\|^2 \le 2\tau^2\sigma_*^2 + 4\tau^2 L(f(\mathbf{x}_t) - f(\mathbf{x}^*)). \tag{2.7}$$

The term $f(\mathbf{x}_t) - f(\mathbf{x}^\star)$ on the right-hand side has the same form as the descent term in (2.5). For $\tau \le 1/(2L)$ and $\mathbf{z} = \mathbf{x}^*$ for some $\mathbf{x}^*$, the variance contribution $4\tau^2 L(f(\mathbf{x}_t) - f(\mathbf{x}^*))$ is at most $2\tau(f(\mathbf{x}_t) - f(\mathbf{x}^*))$, and thus absorbed by the descent term. The residual $2\tau^2\sigma_*^2$ remains bounded and determines the final convergence rate. This would give the guarantee on the average of the iterates of the algorithm.

For last iterate guarantees, one needs to plug in $\mathbf{z} = \mathbf{z}_t$ for a special choice of $\mathbf{z}_t$ (cf. Section 3.2.2) and apply a more involved analysis (Garrigos et al., 2025). Yet, the mechanism explained above remains key to go beyond the bounded variance or bounded gradient assumptions.

This argument relies on two properties that fail when $g \not\equiv 0$. First, the expansion (2.4) relies on the linear structure of

SGD. The proximal operator is nonlinear, so this identity fails. We replace it with the three-point identity and the prox-inequality (Section 3.2.1). Second, the cancellation $\mathbb{E}[\langle \nabla f_i(\mathbf{x}^\star), \mathbf{x}_t - \mathbf{x}^\star \rangle] = 0$ that is used to get to (2.7) requires $\nabla f(\mathbf{x}^\star) = 0$. With a nontrivial regularizer, denoting $\mathbf{x}^* \in \arg\min h$, the optimality condition $0 \in \nabla f(\mathbf{x}^*) + \partial g(\mathbf{x}^*)$ allows $\nabla f(\mathbf{x}^*) \neq 0$. Overcoming these obstacles while preserving last-iterate guarantees constitutes the main technical contribution of this work.

## 2.2. Related Work

Since we consider two sets of methods, based on stochastic gradient descent or stochastic proximal point methods, we divide the comparison to two sections. A summary of the comparison of our results to the most related ones in the literature is provided in Table 1.

### 2.2.1. SGD WORLD

Most convergence analyses for SGD assume bounded variance (Bottou et al., 2018; Needell et al., 2014; Nemirovski et al., 2009). Several works relax this assumption (Alacaoglu et al., 2025; Gladyshev, 1965; Khaled & Richtárik, 2023; Khaled et al., 2023; Moulines & Bach, 2011; Neu & Okolo, 2024; Poljak & Tsypkin, 1973), though most focus on the average iterate rather than the last iterate, and some are restricted to unconstrained problems.

For the last iterate, most results assume bounded variance, bounded gradients, or other distributional assumptions on the gradient noise (Harvey et al., 2019; Liu & Zhou, 2024; Orabona, 2020; Shamir & Zhang, 2013); some extend to composite problems. The influential work by Moulines & Bach (2011) established a suboptimal $O(T^{-1/3})$ rate for the last iterate. Two recent works (Attia et al., 2025; Garrigos et al., 2025) achieve the near-optimal $\widetilde{O}(T^{-1/2})$ rate without bounded variance, under (1.2).

A key feature of these results is *N-independence*: they apply even when the number of component functions $N$ in the sum $f(\mathbf{x}) = (1/N)\sum_{i=1}^{N} f_i(\mathbf{x})$ is infinite, with convergence bounds that do not depend on $N$.

A complementary line of work focuses on finite $N$ with non-i.i.d. sampling, such as without replacement or cyclic selection rather than with replacement. Two recent works establish last-iterate guarantees: Cai & Diakonikolas (2025) study incremental gradient and incremental proximal-point methods for unconstrained problems, where components are selected cyclically; Liu & Zhou (2025) address the composite problem but require $\text{prox}_g$.

*The key distinction is the following:* these convergence bounds depend polynomially on $N$, hence require a finite number of component functions; our bounds are $N$-independent. That is, the results of Cai & Diakonikolas

(2025) and Liu & Zhou (2025) require multiple passes over the data for making progress, whereas our bounds show progress with a single pass. On the other hand, if one is to focus on this different setting with finite $N$, these results may give a better complexity. Moreover, neither work considers stochastic sampling of regularizers (using $\text{prox}_{g_i}$ when $g(\mathbf{x}) = \sum_{i=1}^{m} g_i(\mathbf{x})$), which is the focus of Sec. 4.

### 2.2.2. PROXIMAL POINT WORLD

We now consider the more general setting where the algorithm accesses unbiased samples of both the smooth part $f$ and the nonsmooth part $g$. In particular, given a finite-sum problem

$$\min_{\mathbf{x}} \frac{1}{N}\sum_{i=1}^{N} f_i(\mathbf{x}) + g_i(\mathbf{x}), \tag{2.8}$$

stochastic proximal point (SPP) methods iterate as

$$\mathbf{x}_{t+1} = \text{prox}_{\tau(f_i+g_i)}(\mathbf{x}_t),$$

for $i \in \{1, \ldots, N\}$ selected uniformly at random.

The proximal operator $\text{prox}_{f_i+g_i}$ is often intractable even when $\text{prox}_{f_i}$ and $\text{prox}_{g_i}$ admit closed forms. Consequently, most works on stochastic proximal point methods assume $g_i = 0$ (Asi & Duchi, 2019; Bianchi, 2016). A more practical approach treats the smooth and nonsmooth terms separately, and apply the gradient of $f_i$ followed by the proximal operator of $g_i$, see (RIPM). Bertsekas (2011) analyzed this strategy in order to establish asymptotic convergence of the sum $f(\mathbf{x}_k) + g(\mathbf{x}_k)$ but without nonasymptotic rates for the last iterate.

Recent work by Tovmasyan et al. (2025) and Condat et al. (2025) provides nonasymptotic guarantees for related methods, but under strong convexity assumptions that exclude many regularized problems of practical interest.

Variance reduction techniques have also been applied to stochastic proximal point methods. Traoré et al. (2024) study finite sum problems and establish an ergodic rate of $\mathcal{O}(1/T)$ for convex smooth objectives. However, their focus is on the averaged iterate rather than the last iterate, and they do not consider the case where $g$ is accessed via $\text{prox}_{g_i}$.

## 3. Proximal SGD

This section establishes the last-iterate convergence rate for proximal SGD (SPGD). We state the main theorem, outline the proof strategy, and specialize the result to projected SGD. Full proofs appear in Appendix A.

### 3.1. Statement of the result

The following theorem gives the last-iterate convergence rate for (SPGD). The proof appears in Appendix A.4.

| | Oracle for $g$ | Unbounded variance and no distribution assumption? | Handles prox? | Last iter. | Bound indep. # smooth funcs | Stochastic prox |
|---|---|---|---|---|---|---|
| Folklore | $\text{prox}_g$ | × | ✓ | × | ✓ | × |
| Khaled et al. (2023) | $\text{prox}_g$ | ✓ | ✓ | × | ✓ | × |
| Attia et al. (2025); Garrigos et al. (2025) | N/A | ✓ | × | ✓ | ✓ | × |
| Liu & Zhou (2024) | $\text{prox}_g$ | × | ✓ | ✓ | ✓ | × |
| Bertsekas (2011) | $\text{prox}_{g_i}$ | × | ✓ | ✓ | ✓ | ✓ |
| Section 3 | $\text{prox}_g$ | ✓ | ✓ | ✓ | ✓ | × |
| Section 4 | $\text{prox}_{g_i}$ | ✓ | ✓ | ✓ | ✓ | ✓ |

*Table 1.* Comparison of relevant results

**Theorem 3.1** (Last-iterate convergence)**.** *Let Assumption 2.1 hold. In* (SPGD)*, let* $\tau = 1/(3L\sqrt{T})$*. Then*

$$\mathbb{E}\left[h(\mathbf{x}_{T+1}) - h^*\right] \leq \frac{9}{\sqrt{T}}\Big[LD_*^2 + \frac{1}{\sqrt{T}}(h(\mathbf{x}_0) - h^*)$$
$$+ \frac{\sigma_*^2}{L}\Big(\frac{1}{T} + 4\ln(T+1)\Big)\Big].$$

The $O(\ln(T+1)/\sqrt{T})$ rate matches known last-iterate guarantees for unconstrained SGD (Attia et al., 2025; Garrigos et al., 2025). Compared to the optimal $O(1/\sqrt{T})$ rate that is achieved by averaged iterates or with non-standard step sizes under bounded variance (Jain et al., 2019), our bound incurs an extra logarithmic term of $\ln(T+1)$.

When the objective has finite-sum structure $f = \frac{1}{N}\sum_{i=1}^N f_i$, our bound is independent of $N$, which allows $N$ to be arbitrarily large and is a key benefit of SGD, see Section 2.2.1 for this discussion. The bound depends only on the smoothness constant $L$, the variance at the solution $\sigma_*^2$, and the squared initial distance $D_*^2$; see Section 2 for precise definitions.

### 3.2. Proof setup

The proof of Theorem 3.1 proceeds in two stages: a one-iteration analysis that bounds the expected progress per step, followed by a last-iterate reduction that telescopes these bounds into a final iterate guarantee.

#### 3.2.1. ONE-ITERATION ANALYSIS

The essential intermediate result is a one-iteration bound for proximal SGD that generalizes Garrigos et al. (2025, Lemma 4.2) to the proximal setting. The full proof appears in Appendix A.1.

**Lemma 3.2** (Per-iteration progress)**.** *Let Assumption 2.1 hold. In* (SPGD)*, pick* $\tau$ *such that* $\tau < 1/(2L)$*. Then for all*

$t \in [0, T]$ *and* $\mathbf{z}_t$ *in* $\mathcal{F}(\mathbf{x}_0, ..., \mathbf{x}_t)$ *it holds that*

$$\mathbb{E}\left[h(\mathbf{x}_{t+1}) - h(\mathbf{z}_t) - \mu h(\mathbf{x}_t) + \mu h^*\right]$$
$$\leq \frac{1}{2\tau}\mathbb{E}\left[\|\mathbf{x}_t - \mathbf{z}_t\|^2 - \|\mathbf{x}_{t+1} - \mathbf{z}_t\|^2\right] + v,$$

*where* $\mu = 2\tau L(1+\gamma)$ *and* $v = \left(1 + \frac{1}{\gamma}\right)\sigma_*^2\tau$ *for any* $\gamma > 0$*.*

Unlike Garrigos et al. (2025, Lemma 4.2), the left-hand side depends not only on $\mathbf{x}_t$ and $\mathbf{z}_t$ but also on $\mathbf{x}_{t+1}$. Lemma 3.3 below shows that the technique of Zamani & Glineur (2025) accommodates this modification.

We now sketch the proof and highlight the key differences from the unconstrained case.

As discussed in Section 2.1, the main difficulty is the loss of linearity between $\mathbf{x}_{t+1}$ and $\mathbf{x}_t$. Instead of the expansion (2.4), we now have

$$\|\mathbf{x}_{t+1} - \mathbf{z}\|^2 = \|\mathbf{x}_t - \mathbf{z}\|^2 + 2\langle\mathbf{x}_{t+1} - \mathbf{x}_t, \mathbf{x}_t - \mathbf{z}\rangle \\ + \|\mathbf{x}_{t+1} - \mathbf{x}_t\|^2, \quad (3.1)$$

which is expanding the squared norm. The second term on the right-hand side requires a bound.

The standard analysis of proximal SGD relies on the proximal inequality, which follows from the definition of the proximal operator and convexity of $g$ (see Lemma A.3):

$$\langle\mathbf{x}_{t+1} - \mathbf{x}_t + \tau\nabla f_{i_t}(\mathbf{x}_t), \mathbf{z} - \mathbf{x}_{t+1}\rangle \\ \geq \tau(g(\mathbf{x}_{t+1}) - g(\mathbf{z})). \quad (3.2)$$

Applying this inequality to the second term in (3.1) after adding and subtracting $\langle\mathbf{x}_{t+1} - \mathbf{x}_t, \mathbf{x}_{t+1}\rangle$ yields

$$\|\mathbf{x}_{t+1} - \mathbf{z}\|^2 - \|\mathbf{x}_t - \mathbf{z}\|^2 \\ \leq -\|\mathbf{x}_{t+1} - \mathbf{x}_t\|^2 + 2\tau(g(\mathbf{z}) - g(\mathbf{x}_{t+1})) \quad (3.3) \\ + 2\tau\langle\nabla f_{i_t}(\mathbf{x}_t), \mathbf{z} - \mathbf{x}_{t+1}\rangle.$$

The error term $\langle \nabla f_{i_t}(\mathbf{x}_t), \mathbf{z} - \mathbf{x}_{t+1} \rangle$ resembles the corresponding term in (2.4), but with $\mathbf{x}_t$ in the gradient and $\mathbf{x}_{t+1}$ in the inner product. This mismatch means we cannot directly apply convexity. Additional estimation via smoothness is required; see Appendix A.1. Setting $\mathbf{z} = \mathbf{z}_t$, as defined below, completes the proof.

### 3.2.2. LAST-ITERATE REDUCTION

The remainder of the analysis utilizes ideas from Zamani & Glineur (2025), by also using Garrigos et al. (2025). The next lemma is stated abstractly because it also applies to the incremental proximal method in Section 4.

Following these works, we define the auxiliary sequence $\mathbf{z}_t = (1 - p_t)\mathbf{x}_t + p_t \mathbf{z}_{t-1}$ for $t \geq 0$ with $p_0 = 1$, $\mathbf{z}_{-1} = \mathbf{x}^*$, and, for $t \geq 1$,

$$p_t = \frac{\mu + T - t + 1}{T - t + 2}.$$

The key insight of Zamani & Glineur (2025) is that this convex combination, with $\mathbf{z}_{-1} = \mathbf{x}^*$, leads us to last-iterate guarantees. Note that $p_t \in [0, 1]$ because $\mu < 1$ (where $\mu$ is defined in Lemma 3.2), which holds when $\gamma = \frac{1 - 2\tau L}{1 + 2\tau L}$ and $\tau < \frac{1}{2L}$ which is enforced in our step size rule. We now apply this idea to our setting; the proof appears in Appendix A.2.

**Lemma 3.3** (Last iterate reduction). *Let $h$ be convex. Suppose that for all $t \in [0, T]$, the inequality*

$$\mathbb{E}[h(\mathbf{x}_{t+1}) - h(\mathbf{z}_t) - \mu h(\mathbf{x}_t) + \mu h^*]$$
$$\leq \frac{1}{2\tau}\mathbb{E}[\|\mathbf{x}_t - \mathbf{z}_t\|^2 - \|\mathbf{x}_{t+1} - \mathbf{z}_t\|^2] + \mathcal{E} \quad (3.4)$$

*holds for some constant $\mathcal{E}$. Then*

$$\alpha_T \mathbb{E}[h(\mathbf{x}_{T+1}) - h^*] \leq \frac{1}{2\tau}\|\mathbf{x}_0 - \mathbf{x}^*\|^2 + h(\mathbf{x}_0) - h^*$$
$$+ \mathcal{E}\left(\sum_{t=1}^{T} \alpha_t + \alpha_0\right), \quad (3.5)$$

*where the sequence $(\alpha_t)$ is defined by $\alpha_{-1} = \alpha_0 = 1$ and*

$$\alpha_t = \frac{T - t + 2}{\mu + T - t + 1} \cdot \alpha_{t-1}, \qquad t = 1, \ldots, T.$$

From the definition of $\alpha_t$, we have $\alpha_t p_t = \alpha_{t-1}$ for all $t \geq 0$, so $\alpha_t$ is nondecreasing, since $p_t \in [0, 1]$. The convergence rate depends on the growth of $\alpha_t$, which is analyzed in App. A.3 using ideas from Garrigos et al. (2025); Zamani & Glineur (2025).

Unlike Garrigos et al. (2025), our recursion involves both $h(\mathbf{x}_{t+1})$ and $h(\mathbf{x}_t)$ due to the existence of the proximal operator in the method. We show that the technique of Zamani & Glineur (2025) extend naturally to this setting.

Combine Lemmas 3.2 and 3.3 with the bounds on $\alpha_t$ from Appendix A.3 to obtain Theorem 3.1.

### 3.3. Corollary for Projected SGD

Theorem 3.1 specializes to projected SGD over a closed convex set $C \subset \mathbb{R}^n$ by setting $g = \delta_C$, where the indicator function $\delta_C(\mathbf{x}) = 0$ if $\mathbf{x} \in C$ and $\delta_C(\mathbf{x}) = +\infty$ otherwise. Since $\delta_C$ is proper, convex, and lower semicontinuous, Assumption 2.1 holds. The problem becomes

$$\min_{\mathbf{x} \in C} f(\mathbf{x}),$$

and the iteration (SPGD) reduces to

$$\mathbf{x}_{t+1} = \text{proj}_C(\mathbf{x}_t - \tau \nabla f_{i_t}(\mathbf{x}_t)), \qquad \text{(projSGD)}$$

where $\text{proj}_C(\mathbf{x}) = \arg\min_{\mathbf{y} \in C} \|\mathbf{y} - \mathbf{x}\|$ is the Euclidean projection onto $C$.

**Corollary 3.4** (Projected SGD). *Let Assumption 2.1 hold for $f$, and let $C$ be convex and closed. For (projSGD), let $\tau = 1/(3L\sqrt{T})$. Then,*

$$\mathbb{E}[f(\mathbf{x}_{T+1}) - f^*] = O\left(\frac{\ln(T + 1)}{\sqrt{T}}\right).$$

We emphasize that the results of the form above for the last iterate under Assumption 2.1 were only known for SGD for unconstrained problems prior to our work. Since the statement is a special case of Theorem 3.1, its proof is omitted.

## 4. Randomized Incremental Proximal Method

This section extends our analysis to additive regularizers of the form $g = \sum_{i=1}^{m} g_i$, where we access only the proximal operators of the component functions $g_i$. The standard algorithm for this setting is the randomized incremental proximal method (RIPM), introduced by Bertsekas (2011).

Although this setting generalizes (SPGD), the analysis requires an additional assumption: each $g_i$ must be Lipschitz continuous (Assumption 2.2). This assumption is standard when proximal operators are accessed stochastically (Bertsekas, 2011).

We state the main result below, followed by key proof ideas and a corollary. We then examine why this Lipschitzness assumption is necessary compared to the setting of Section 3. Complete proofs appear in Appendix B.

### 4.1. Statement of the result

The following theorem establishes a last-iterate convergence rate for (RIPM). The proof appears in Appendix B.2.

**Theorem 4.1** (Last-iterate convergence, incremental proximal). *Let Assumptions 2.1 and 2.2 hold. For (RIPM), set*

$\tau = 1/(5L\sqrt{T})$. *Then*

$$\mathbb{E}[h(\mathbf{x}_{T+1}) - h^*] \leq \frac{10}{\sqrt{T}}\Big[LD_*^2 + \frac{1}{\sqrt{T}}(h(\mathbf{x}_0) - h^*) \\ + \frac{\sigma_*^2 + 4m^2L_g^2}{L}\Big(\frac{1}{T} + 4\ln(T+1)\Big)\Big].$$

This result extends Theorem 3.1 to handle stochastic access to the regularizer $g$, and requires only the individual proximal operators $\text{prox}_{g_j}$ rather than $\text{prox}_g$. The generalization incurs an additional variance term $4m^2L_g^2$ in the bound and the Lipschitz requirement on each $g_j$; see Assumption 2.2. As in Theorem 3.1, the smooth component $f$ does not require a finite-sum structure.

The most closely related result is Bertsekas (2011), who assumed uniformly bounded gradients $\|\nabla f_i(\mathbf{x})\|$ and proved only asymptotic convergence without explicit rates.

### 4.2. Proof setup

We sketch the proof and justify the Lipschitzness assumption on $g_j$, following the framework of Section 3.2.

Instead of (3.2), we now have

$$\langle \mathbf{x}_{t+1} - \mathbf{x}_t + \tau\nabla f_{i_t}(\mathbf{x}_t), \mathbf{z} - \mathbf{x}_{t+1}\rangle \\ \geq \tau m(g_{j_t}(\mathbf{x}_{t+1}) - g_{j_t}(\mathbf{z})). \tag{4.1}$$

The aim is to combine this inequality with (3.1) to obtain a recursion analogous to (3.3). The main difference from Section 3.2 is that we have $m(g_{j_t}(\mathbf{x}_{t+1}) - g_{j_t}(\mathbf{z}))$ instead of $g(\mathbf{x}_{t+1}) - g(\mathbf{z})$. The difficulty becomes apparent:

$$\mathbb{E}_t[m(g_{j_t}(\mathbf{x}_{t+1}) - g_{j_t}(\mathbf{z}))] \neq g(\mathbf{x}_{t+1}) - g(\mathbf{z}).$$

The iterate $\mathbf{x}_{t+1}$ depends on $j_t$, so $g_{j_t}$ and $\mathbf{x}_{t+1}$ are coupled and the conditional expectation does not factor; see (RIPM). The resolution is to decouple them using Lipschitzness:

$$mg_{j_t}(\mathbf{x}_{t+1}) = mg_{j_t}(\mathbf{x}_t) + m(g_{j_t}(\mathbf{x}_{t+1}) - g_{j_t}(\mathbf{x}_t)) \\ \leq mg_{j_t}(\mathbf{x}_t) + mL_g\|\mathbf{x}_{t+1} - \mathbf{x}_t\|.$$

With $g_{j_t}$ evaluated at $\mathbf{x}_t$, which is $\mathcal{F}_t$-measurable, we have $\mathbb{E}_t[mg_{j_t}(\mathbf{x}_t)] = g(\mathbf{x}_t)$. A second application of Lipschitzness then recovers $g(\mathbf{x}_{t+1})$. This is the standard argument for handling such coupling (Bertsekas, 2011).

This decoupling introduces additional error terms on the right-hand side of the one-iteration bound, compared to Lemma 3.2. See App. B.1 for the proof of the next lemma.

**Lemma 4.2** (Per-iteration descent, incremental proximal). *Let Assumptions 2.1 and 2.2 hold. For* (RIPM), *set $\tau < 1/(4L)$. Then for all $t = 0, \ldots, T$ and $\mathbf{z}_t \in \mathcal{F}(\mathbf{x}_0, \ldots, \mathbf{x}_t)$ it holds that*

$$\mathbb{E}_t\left[h(\mathbf{x}_{t+1}) - h(\mathbf{z}_t) - \mu h(\mathbf{x}_t) + \mu h^*\right] \\ \leq \frac{1}{2\tau}\mathbb{E}_t\left[\|\mathbf{x}_t - \mathbf{z}_t\|^2 - \|\mathbf{x}_{t+1} - \mathbf{z}_t\|^2\right] + v + 8\tau m^2 L_g^2,$$

*where $\mu = 2\tau L(1 + \gamma)$ and $v = \left(1 + \frac{1}{\gamma}\right)\sigma_*^2\tau$ for any $\gamma > 0$.*

Applying Lemma 3.3 with $\mathcal{E} = v + 8\tau m^2 L_g^2$ completes the proof of Theorem 4.1.

### 4.3. Corollary for Stochastic Proximal Point Method

The stochastic proximal point (SPP) method applies to (1.1) when $f \equiv 0$. Given an objective $g(\mathbf{x}) := \frac{1}{m}\sum_{j=1}^m g_j(\mathbf{x})$, the method generates iterates

$$\mathbf{x}_{t+1} = \text{prox}_{\tau g_{j_t}}(\mathbf{x}_t),$$

where $j_t$ is sampled uniformly at random from $\{1, \ldots, m\}$.

Though well-studied, we are not aware of an optimal last-iterate rate guarantee for this method under mere convexity. The proof of the next result is given in Appendix B.3.

**Corollary 4.3** (Stochastic proximal point). *Let Assumption 2.2 hold. For the stochastic proximal point method, set $\tau = 1/(\sqrt{T+1})$. Then, we have*

$$\mathbb{E}[g(\mathbf{x}_{T+1}) - g(\mathbf{x}^*)] = O\left(\frac{\ln(T+1)}{\sqrt{T}}\right).$$

A suboptimal $O(T^{-1/3})$ rate was shown for this method in (Toulis et al., 2021). The other closest prior result is Cai & Diakonikolas (2025, Section 3.2), which analyzes *cyclic* incremental proximal point for finite-sum objectives $\frac{1}{N}\sum_{i=1}^N g_i(\mathbf{x})$. To enable direct comparison, we apply our analysis to the same normalized formulation. Our rate improves upon theirs by a factor of $\sqrt{N}$. We analyze SPP with i.i.d. sampling rather than cyclic selection. Their bound is deterministic whereas ours holds in expectation, making the results complementary.

### 4.4. Extensions to BlockProx

We now apply the incremental proximal framework to BlockProx, a distributed algorithm for graph-structured regularizers. Consider the optimization problem with partially separable objectives

$$\min_{\mathbf{x}} \, h(\mathbf{x}) := \sum_{i=1}^N f_i(\mathbf{x}_i) + \sum_{j=1}^m g_j(\mathbf{x}),$$

where each $\mathbf{x}_i \in \mathbb{R}^d$ denotes the variables at node $i$ and $\mathbf{x} = (\mathbf{x}_1, \ldots, \mathbf{x}_N) \in \mathbb{R}^{Nd}$ joins all node variables. This matches our framework with $f(\mathbf{x}) = \sum_{i=1}^N f_i(\mathbf{x}_i)$ and $g(\mathbf{x}) = \sum_{j=1}^m g_j(\mathbf{x})$, where each $f_i: \mathbb{R}^d \to \mathbb{R}$ is convex and smooth, and each $g_j$ is convex and $L_g$-Lipschitz. For each regularizer $g_j$, let

$$S_j \subseteq \{1, \ldots, N\}$$

denote the support of $g_j$, namely the set of nodes that $g_j$ depends on. That is, $g_j(\mathbf{x}) = g_j(\mathbf{x}_{S_j})$ where $\mathbf{x}_{S_j} = (\mathbf{x}_i \mid i \in S_j)$.

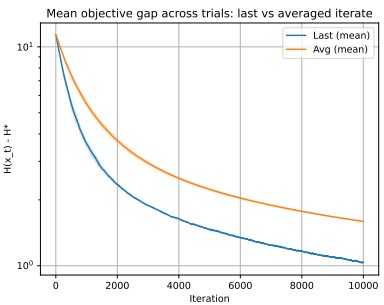
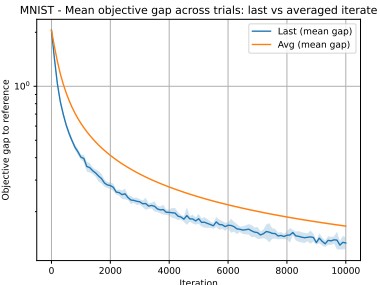

*(a)* Synthetic Lasso objective gap $H(\mathbf{x}_t) - H^*$ (semilog scale).

*(b)* MNIST logistic regression: objective gap to reference solution (semilog scale).

*Figure 1.* Comparison of last and averaged iterates.

In particular, $m$ may equal the number of edges in a graph, with $g_j$ coordinating communication between nodes sharing edge $j$.

The gradient $\nabla f(\mathbf{x}) = (\nabla f_1(\mathbf{x}^{(1)}), \ldots, \nabla f_N(\mathbf{x}^{(N)}))$ decomposes by separability. The BlockProx algorithm of Lin et al. (2025) iterates as

$$\mathbf{y}_t = \mathbf{x}_t - \tau \nabla f(\mathbf{x}_t),$$
$$\mathbf{x}_{t+1}^{(i)} = \left[\text{prox}_{m\tau g_{j_{i,t}}}(\mathbf{y}_t)\right]_i \in \mathbb{R}^d, \text{ if } i \in S_{j_{i,t}}, \quad (4.2)$$
$$\mathbf{x}_{t+1}^{(i)} = \mathbf{y}_t^{(i)}, \text{ if } i \notin S_{j_{i,t}},$$

where $j_{i,t}$ is drawn uniformly at random from $1, \ldots, m$, which is the edge that node $i$ selects at iteration $t$. All the nodes are allowed to sample another edge and this process runs over all the nodes in parallel.

The analysis in Lin et al. (2025) reduces BlockProx to RIPM without stochastic gradients. Specifically, Proposition 5.2 of that work translates convergence results for RIPM to BlockProx.

In particular, that work showed BlockProx achieves $\widetilde{O}(1/\sqrt{T})$ convergence either *(i)* on the last iterate when the gradients of $f$ are bounded (Lin et al., 2025, Thm. 5.6), or *(ii)* on the average iterate when $f$ is smooth (Lin et al., 2025, Thm. 5.9). However, the experiments therein use linear least squares for $f$ and output the last iterate. Hence, the existing analysis does not cover the method as implemented. We bridge this theory-practice gap by proving last-iterate rates when each $f_i$ is convex and smooth, without bounded gradients. The proof of the following theorem appears in Appendix B.4.

**Theorem 4.4** (Last-iterate convergence, BlockProx). *Let Assumptions 2.1 and 2.2 hold. Consider the iterates of BlockProx with $\tau = 1/(3L\sqrt{T})$. Then, we have*

$$\mathbb{E}[h(\mathbf{x}_T) - h^*] = O\left(\frac{\ln(T+1)}{\sqrt{T}}\right).$$

Although RIPM generalizes BlockProx by including stochastic gradients, BlockProx uses deterministic gradients of $f$. This simpler structure admits a direct proof: we adapt the one-iteration analysis in Lin et al. (2025) to the framework of Garrigos et al. (2025).

## 5. Numerical Experiments

Practitioners routinely output the last iterate of SGD-based methods rather than the average. We provide experiments that complement our theoretical analysis to show that this common practice is indeed useful.

We compare the last iterate $\mathbf{x}_T$ and the averaged iterate $\bar{\mathbf{x}}_T = (1/T) \sum_{t=1}^{T} \mathbf{x}_t$ on two problems: Lasso regression with synthetic data and $\ell_1$-regularized logistic regression on MNIST. All results are averaged over 10 independent trials with fixed datasets.

**Synthetic Lasso.** We solve

$$\min_{\mathbf{x}} \left\{ h(\mathbf{x}) = \frac{1}{2n}\|A\mathbf{x} - \mathbf{y}\|_2^2 + \lambda\|\mathbf{x}\|_1 \right\},$$

using a synthetic matrix $A$ and sparse ground truth. We compute the optimal value $h^*$ using CVXPY and plot the optimality gap $h(\mathbf{x}_t) - h^*$ over iterations. The stepsize is $\tau = 1/(4L\sqrt{T})$, consistent with our theoretical prescription. Figure 1a shows that while the averaged iterate decreases more smoothly, the last iterate achieves a smaller optimality gap across all trials.

**$\ell_1$-Regularized Logistic Regression.** We solve multiclass logistic regression with $\ell_1$ regularization on MNIST using (SPGD) with stepsize $\tau = \Theta(1/\sqrt{T})$. That is,

$$\min_{\mathbf{X} \in \mathbb{R}^{d \times K}} \left\{ h(\mathbf{X}) = \frac{1}{n}\sum_{i=1}^{n} \ell(\mathbf{X}; a_i, y_i) + \lambda\|\mathbf{X}\|_1 \right\},$$

where

$$\ell(\mathbf{X}; a_i, y_i) = -\langle \mathbf{X}_{y_i}, a_i \rangle + \log \sum_{k=1}^{K} e^{\langle \mathbf{X}_k, a_i \rangle}$$

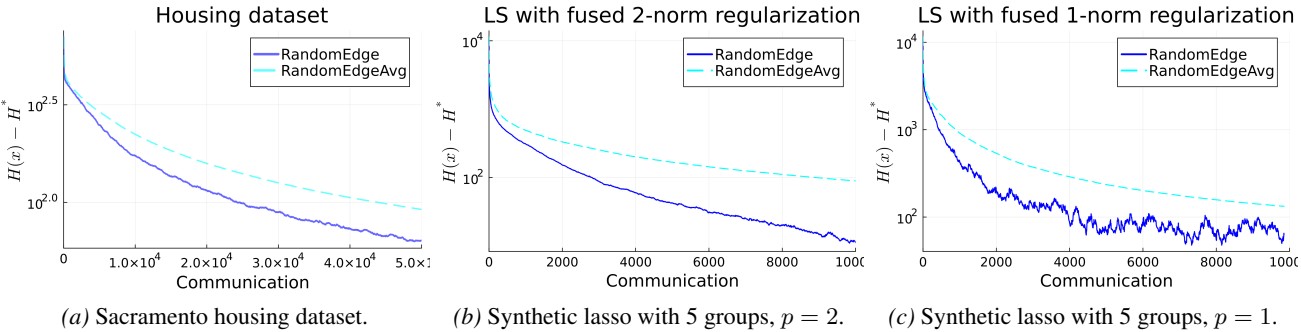

*(a)* Sacramento housing dataset.   *(b)* Synthetic lasso with 5 groups, $p = 2$.   *(c)* Synthetic lasso with 5 groups, $p = 1$.

*Figure 2.* Comparison of last and averaged iterates over communication rounds for BlockProx.

is the multiclass logistic loss, $a_i \in \mathbb{R}^d$ is the $i$-th data point, $y_i \in \{1, \ldots, K\}$ is its label and $\mathbf{X}_{y_i}$ is the weight vector corresponding to class $y_i$. For MNIST, $d = 784$ and $K = 10$. We compute a reference solution $h^*$ using the SAGA solver from Scikit-Learn (Pedregosa et al., 2011). Figure 1b confirms that the last iterate typically achieves a smaller optimality gap than the averaged iterate, consistent with the Lasso experiment.

**Graph-structured regularization (BlockProx).** We solve a graph-structured optimization problem using the BlockProx algorithm of Lin et al. (2025). Given a graph $\mathcal{G} = (\mathcal{V}, \mathcal{E})$ with $|\mathcal{V}| = n$ and the edges $\mathcal{E}$ capture the communication structure, we solve

$$\min_{\mathbf{x}} \left\{ h(\mathbf{x}) = \sum_{i \in \mathcal{V}} f_i(\mathbf{x}_i) + \sum_{(i,j) \in \mathcal{E}} g_{ij}(\mathbf{x}_i, \mathbf{x}_j) \right\},$$

In particular, we focus on the network lasso setting (Hallac et al., 2015), where

$$f_i(\mathbf{x}_i) = \frac{1}{2} \|A_i \mathbf{x}_i - b_i\|^2, \quad g_{ij}(\mathbf{x}_i, \mathbf{x}_j) = \lambda \|\mathbf{x}_i - \mathbf{x}_j\|_p,$$

where $p \in \{1, 2\}$. For each node $i$, the data matrix $A_i$ is generated either synthetically or from the Sacramento housing dataset (only for $p = 2$) (Hallac et al., 2015). For the synthetic data, we consider three communication network settings:

1. five groups with 10, 17, 18, 18, and 12 nodes, respectively; (see Figure 2 (b) and (c))

2. one group with 20 nodes; (see Figure 3 in Appendix C)

3. one group with 40 nodes, where the communication network is fully connected, i.e., every pair of nodes is connected by an edge. (see Figure 3 in Appendix C)

Additional details regarding the generation of the synthetic data matrix and experimental setup are given in Sections 6.1 and 6.2 of Lin et al. (2025). We compare the last iterate produced by BlockProx with the averaged iterate. Figure 2 shows that the last iterate achieves a smaller optimality gap than the averaged iterate across all settings, consistent with our previous experiments.

## 6. Conclusions and Perspectives

For SPGD and RIPM (that generalize SPP), we proved that the last-iterate has the rate of convergence $\widetilde{O}(1/\sqrt{T})$ with no bounded variance assumptions, under componentwise smoothness and convexity. This rate is $\ln(T + 1)$ away from the optimal rate that is known with bounded variance, and it matches the best-known rates for last-iterate of SGD applied to unconstrained minimization (Attia et al., 2025; Garrigos et al., 2025) or for averaged iterate for problems with unbounded variance (Khaled et al., 2023).

Our work paves the way to many different directions. The first direction is extending our guarantees to methods with random reshuffling or sampling without replacement for selecting stochastic gradients. For this, the techniques developed in Cai & Diakonikolas (2025) and Liu & Zhou (2025) will likely prove useful. The second direction is lifting the Lipschitzness requirement in Section 4. Indeed this assumption is common in the SPP literature. However, for the important special case of $g_i(\mathbf{x}) = \delta_{C_i}(\mathbf{x})$ for a convex and closed set $C_i$, SPP reduces to the well-known method of randomized alternating projections, which admits a separate analysis framework, see for example (Nedić, 2011). Extending these techniques to our template would allow us to solve problems of the form $\min \{f(\mathbf{x}) \mid \mathbf{x} \in \bigcap_{i=1}^m C_i\}$ with last-iterate convergence rates and without bounded variance assumptions on the smooth $f$. Two other directions are extending our analysis to allow Bregman distances or to show high probability guarantees on the last iterate.

## Acknowledgements

Michael P. Friedlander acknowledges the support of the Natural Sciences and Engineering Research Council of Canada (NSERC).

Ahmet Alacaoglu acknowledges the support of the Natural Sciences and Engineering Research Council of Canada (NSERC), [funding reference number RGPIN-2025-06634].

## Impact Statement

"This paper presents work whose goal is to advance the field of Machine Learning. There are many potential societal consequences of our work, none which we feel must be specifically highlighted here."

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

# A. Proofs for Section 3

We will now present our main last-iterate results for (SPGD). For the statement in the main text, that is, Theorem 3.1, we used $C = 3$ and simplified the bounds.

**Theorem A.1** (Last-iterate convergence, proximal SGD, generalized form of Theorem 3.1). *Consider the* (SPGD) *algorithm with step size* $\tau = \frac{1}{CL\sqrt{T}}$ *for some* $C > 2$. *Suppose that assumption 2.1 hold. Then, the last iterate* $\mathbf{x}_{T+1}$ *satisfies*

$$\mathbb{E}\left[h(\mathbf{x}_{T+1}) - h^*\right] \leq \exp\left(\frac{8}{eC}\right)\left[\frac{CL\|\mathbf{x}_0 - \mathbf{x}^*\|^2}{\sqrt{T}} + \frac{2}{T}[h(\mathbf{x}_0) - h^*] + \frac{4\sigma_*^2}{(C-2)LT^{1.5}} + \frac{16\sigma_*^2\ln(T+1)}{(C-2)L\sqrt{T}}\right].$$

We shall now provide the proof of this theorem. Our proof proceeds in four stages:

1. We prove a technical variance transfer lemma that allows us to bound $\mathbb{E}[\|\nabla f_i(\mathbf{x})\|^2]$. See Lemma A.2.

2. We prove a one-step inequality that compares the iterates to a reference point $\mathbf{z}_t \in \mathcal{F}(\mathbf{x}_0, \ldots \mathbf{x}_t)$. See Lemma A.4

3. We use this to prove a bound for the linear combination of objective values including the evaluation at a reference point $\mathbf{z}_t$ which is a convex combination of our iterates. See 3.2.

4. Using the above, we chose the structure of $\mathbf{z}_t$ and weights $\alpha_t$ to obtain a bound for the last iterate.

We start with the variance transfer lemma, which is now classical in the SGD literature, see for example (Khaled et al., 2023, Lemma A.2).

**Lemma A.2** (Variance transfer). *Let Assumption 2.1 hold. Then for every* $\gamma > 0$*, we have*

$$\mathbb{E}_t\|\nabla f_{i_t}(\mathbf{x}_t)\|^2 \leq 2L(1+\gamma)[h(\mathbf{x}_t) - h^*] + \left(1 + \frac{1}{\gamma}\right)\sigma_*^2. \tag{A.1}$$

*Proof.* Let us denote by $\mathbf{x}^*$ an arbitrary solution of our problem. By smoothness and convexity of $f_i$, we have the standard inequality (Nesterov, 2018, Thm. 2.1.5), see also (2.3),

$$\frac{1}{2L}\|\nabla f_{i_t}(\mathbf{x}_t) - \nabla f_{i_t}(\mathbf{x}^*)\|^2 \leq f_{i_t}(\mathbf{x}_t) - f_{i_t}(\mathbf{x}^*) - \langle\nabla f_{i_t}(\mathbf{x}^*), \mathbf{x}_t - \mathbf{x}^*\rangle.$$

We take expectation to get

$$\frac{1}{2L}\mathbb{E}_t\|\nabla f_{i_t}(\mathbf{x}_t) - \nabla f_i(\mathbf{x}^*)\|^2 \leq f(\mathbf{x}_t) - f(\mathbf{x}^*) - \langle\nabla f(\mathbf{x}^*), \mathbf{x}_t - \mathbf{x}^*\rangle. \tag{A.2}$$

From the optimality condition of our problem (1.1), we have that $0 \in \nabla f(\mathbf{x}^*) + \partial g(\mathbf{x}^*)$. Hence, the convexity of $g$ yields

$$g(\mathbf{x}_t) \geq g(\mathbf{x}^*) + \langle-\nabla f(\mathbf{x}^*), \mathbf{x}_t - \mathbf{x}^*\rangle.$$

Substituting into (A.2) gives

$$\frac{1}{2L}\mathbb{E}_t\|\nabla f_{i_t}(\mathbf{x}_t) - \nabla f_{i_t}(\mathbf{x}^*)\|^2 \leq f(\mathbf{x}_t) - f(\mathbf{x}^*) + g(\mathbf{x}_t) - g(\mathbf{x}^*) = h(\mathbf{x}_t) - h^*. \tag{A.3}$$

Here, we use Young's inequality to estimate as

$$\mathbb{E}_t\|\nabla f_{i_t}(\mathbf{x}_t)\|^2 \leq (1+\gamma)\mathbb{E}_t\|\nabla f_{i_t}(\mathbf{x}_t) - \nabla f_{i_t}(\mathbf{x}^*)\|^2 + \left(1 + \frac{1}{\gamma}\right)\mathbb{E}_t\|\nabla f_{i_t}(\mathbf{x}^*)\|^2.$$

We use (A.3) and Assumption 2.1 to bound the first and second terms on the right-hand side, respectively. $\square$

We now restate the optimality condition for the proximal operator that we use in our proofs, which is well known. Its proof follows from the definition of $\mathrm{prox}_{\tau g}$ and the convexity of $g$. See for example (Beck, 2017, Theorem 6.39).

**Lemma A.3** (Prox-inequality). *For any proper, convex, and lower semicontinuous function $g : \mathbb{R}^n \to (-\infty, +\infty]$, we have*

$$\mathbf{u} = \text{prox}_{\tau g}(\mathbf{y}) \iff \langle \mathbf{u} - \mathbf{y}, \mathbf{x} - \mathbf{u} \rangle \geq \tau(g(\mathbf{u}) - g(\mathbf{x})), \quad \text{for all } \mathbf{x} \in \mathbb{R}^n. \tag{A.4}$$

We finally provide a one-step progress inequality with an arbitrary reference point $\mathbf{z}$.

**Lemma A.4** (One-step progress). *Let Assumption 2.1 hold. Then, for the iterates $(\mathbf{x}_t)$ generated by (SPGD), for any $\mathbf{z} \in \mathbb{R}^n$ and $\delta_1 > 0$, it holds that*

$$\|\mathbf{x}_{t+1} - \mathbf{z}_t\|^2 - \|\mathbf{x}_t - \mathbf{z}_t\|^2 \leq 2\tau \left[ f_{i_t}(\mathbf{z}_t) - f_{i_t}(\mathbf{x}_t) + g(\mathbf{z}_t) - g(\mathbf{x}_{t+1}) + f(\mathbf{x}_t) - f(\mathbf{x}_{t+1}) \right]$$
$$+ \frac{\tau}{\delta_1} \|\nabla f_{i_t}(\mathbf{x}_t) - \nabla f(\mathbf{x}_t)\|^2 + (\tau\delta_1 + \tau L - 1)\|\mathbf{x}_t - \mathbf{x}_{t+1}\|^2.$$

*Proof.* Using the standard identity, $\langle \mathbf{a}, \mathbf{b} \rangle = -\frac{1}{2}\|\mathbf{a}\|^2 - \frac{1}{2}\|\mathbf{b}\|^2 + \frac{1}{2}\|\mathbf{a} + \mathbf{b}\|^2$, we have

$$\|\mathbf{x}_{t+1} - \mathbf{z}_t\|^2 - \|\mathbf{x}_t - \mathbf{z}_t\|^2 = \|\mathbf{x}_{t+1} - \mathbf{x}_t\|^2 + 2\langle \mathbf{x}_{t+1} - \mathbf{x}_t, \mathbf{x}_t - \mathbf{z}_t \rangle. \tag{A.5}$$

For the last term on the right-hand side, we use Lemma A.3 with $\mathbf{u} = \mathbf{x}_{t+1}$, $\mathbf{y} = \mathbf{x}_t - \tau\nabla f_{i_t}(\mathbf{x}_t)$, and $\mathbf{x} = \mathbf{z}_t$ to obtain

$$2\langle \mathbf{x}_{t+1} - \mathbf{x}_t, \mathbf{x}_t - \mathbf{z}_t \rangle = 2\langle \mathbf{x}_{t+1} - \mathbf{x}_t, \mathbf{x}_t - \mathbf{x}_{t+1} \rangle + 2\langle \mathbf{x}_{t+1} - \mathbf{x}_t, \mathbf{x}_{t+1} - \mathbf{z}_t \rangle$$
$$\leq -2\|\mathbf{x}_{t+1} - \mathbf{x}_t\|^2 + 2\tau \left( g(\mathbf{z}_t) - g(\mathbf{x}_{t+1}) + \langle \nabla f_{i_t}(\mathbf{x}_t), \mathbf{z}_t - \mathbf{x}_{t+1} \rangle \right).$$

Substituting this into (A.5) gives

$$\|\mathbf{x}_{t+1} - \mathbf{z}_t\|^2 - \|\mathbf{x}_t - \mathbf{z}_t\|^2 \leq -\|\mathbf{x}_{t+1} - \mathbf{x}_t\|^2 + 2\tau(g(\mathbf{z}_t) - g(\mathbf{x}_{t+1})) + 2\tau\langle \nabla f_{i_t}(\mathbf{x}_t), \mathbf{z}_t - \mathbf{x}_{t+1} \rangle. \tag{A.6}$$

We use convexity of $f_{i_t}$ to bound the last term on the right-hand side:

$$2\tau\langle \nabla f_{i_t}(\mathbf{x}_t), \mathbf{z}_t - \mathbf{x}_{t+1} \rangle = 2\tau\langle \nabla f_{i_t}(\mathbf{x}_t), \mathbf{z}_t - \mathbf{x}_t \rangle + 2\tau\langle \nabla f_{i_t}(\mathbf{x}_t), \mathbf{x}_t - \mathbf{x}_{t+1} \rangle$$
$$\leq 2\tau[f_{i_t}(\mathbf{z}_t) - f_{i_t}(\mathbf{x}_t)] + 2\tau\langle \nabla f_{i_t}(\mathbf{x}_t), \mathbf{x}_t - \mathbf{x}_{t+1} \rangle. \tag{A.7}$$

Adding and subtracting $\nabla f(\mathbf{x}_t)$ in the inner product gives us:

$$2\tau\langle \nabla f_{i_t}(\mathbf{x}_t), \mathbf{x}_t - \mathbf{x}_{t+1} \rangle = 2\tau\langle \nabla f_{i_t}(\mathbf{x}_t) - \nabla f(\mathbf{x}_t), \mathbf{x}_t - \mathbf{x}_{t+1} \rangle + 2\tau\langle \nabla f(\mathbf{x}_t), \mathbf{x}_t - \mathbf{x}_{t+1} \rangle. \tag{A.8}$$

We estimate the first term on the right-hand side by using the Young's inequality:

$$2\tau\langle \nabla f_{i_t}(\mathbf{x}_t) - \nabla f(\mathbf{x}_t), \mathbf{x}_t - \mathbf{x}_{t+1} \rangle \leq \frac{\tau}{\delta_1}\|\nabla f_{i_t}(\mathbf{x}_t) - \nabla f(\mathbf{x}_t)\|^2 + \tau\delta_1\|\mathbf{x}_t - \mathbf{x}_{t+1}\|^2.$$

For the second term on the right-hand side of (A.8), we can use the smoothness of $f$ to derive

$$2\tau\langle \nabla f(\mathbf{x}_t), \mathbf{x}_t - \mathbf{x}_{t+1} \rangle \leq 2\tau\left( f(\mathbf{x}_t) - f(\mathbf{x}_{t+1}) + \frac{L}{2}\|\mathbf{x}_t - \mathbf{x}_{t+1}\|^2 \right).$$

We plug in the last two estimates to (A.8) which we then insert in (A.7) to get the estimate

$$2\tau\langle \nabla f_{i_t}(\mathbf{x}_t), \mathbf{z}_t - \mathbf{x}_{t+1} \rangle \leq 2\tau[f_{i_t}(\mathbf{z}_t) - f_{i_t}(\mathbf{x}_t)] + \frac{\tau}{\delta_1}\|\nabla f_{i_t}(\mathbf{x}_t) - \nabla f(\mathbf{x}_t)\|^2$$
$$+ \tau\delta_1\|\mathbf{x}_t - \mathbf{x}_{t+1}\|^2 + 2\tau\left( f(\mathbf{x}_t) - f(\mathbf{x}_{t+1}) + \frac{L}{2}\|\mathbf{x}_t - \mathbf{x}_{t+1}\|^2 \right).$$

Using this bound in (A.6) and rouping the terms involving $\|\mathbf{x}_t - \mathbf{x}_{t+1}\|^2$ gives the assertion. $\square$

### A.1. Bounding a linear combination of function values

We now prove Lemma 3.2. We restate it here for the reader's convenience.

**Lemma A.5** (Per-iteration progress, see Lemma 3.2). *Let Assumption 2.1 hold. In* (SPGD), *pick $\tau$ such that $\tau < 1/(2L)$. Then for all $t \in [0, T]$ and $\mathbf{z}_t$ in $\mathcal{F}(\mathbf{x}_0, ..., \mathbf{x}_t)$ it holds that*

$$\mathbb{E}\left[h(\mathbf{x}_{t+1}) - h(\mathbf{z}_t) - \mu h(\mathbf{x}_t) + \mu h^*\right] \leq \frac{1}{2\tau}\mathbb{E}\left[\|\mathbf{x}_t - \mathbf{z}_t\|^2 - \|\mathbf{x}_{t+1} - \mathbf{z}_t\|^2\right] + \tau\left(1 + \frac{1}{\gamma}\right)\sigma_*^2,$$

*where $\mu = 2\tau L(1+\gamma)$ for any $\gamma > 0$.*

*Proof.* On the result of Lemma A.4, we take conditional expectation and rearrange (noting the cancellation of the terms $\mathbb{E}_t[f_{i_t}(\mathbf{x}_t)]$ and $f(\mathbf{x}_t)$) to derive

$$2\tau\mathbb{E}_t[f(\mathbf{x}_{t+1}) + g(\mathbf{x}_{t+1}) - f(\mathbf{z}_t) - g(\mathbf{z}_t)] \leq \|\mathbf{x}_t - \mathbf{z}_t\|^2 - \mathbb{E}_t\|\mathbf{x}_{t+1} - \mathbf{z}_t\|^2 + \frac{\tau}{\delta_1}\mathbb{E}_t\|\nabla f_{i_t}(\mathbf{x}_t) - \nabla f(\mathbf{x}_t)\|^2$$
$$+ (\tau\delta_1 + \tau L - 1)\mathbb{E}_t\|\mathbf{x}_t - \mathbf{x}_{t+1}\|^2.$$

We next use the bound $\mathbb{E}_t\|\nabla f_{i_t}(\mathbf{x}_t) - \nabla f(\mathbf{x}_t)\|^2 \leq \mathbb{E}_t\|\nabla f_{i_t}(\mathbf{x}_t)\|^2$ and Lemma A.2 to get

$$2\tau\mathbb{E}_t[f(\mathbf{x}_{t+1}) + g(\mathbf{x}_{t+1}) - f(\mathbf{z}_t) - g(\mathbf{z}_t)] \leq \|\mathbf{x}_t - \mathbf{z}_t\|^2 - \mathbb{E}_t\|\mathbf{x}_{t+1} - \mathbf{z}_t\|^2$$
$$+ \frac{\tau}{\delta_1}\left(2L(1+\gamma)[h(\mathbf{x}_t) - h^*] + \left(1 + \frac{1}{\gamma}\right)\sigma_*^2\right)$$
$$+ (\tau\delta_1 + \tau L - 1)\mathbb{E}_t\|\mathbf{x}_t - \mathbf{x}_{t+1}\|^2. \tag{A.9}$$

We now set $\delta_1 = \frac{1}{\tau} - L > 0$ so that the last term in (A.9) becomes nonpositive. We next use $\frac{\tau}{\delta_1} = \frac{\tau^2}{1-\tau L} \leq 2\tau^2$, which is by $\tau L < \frac{1}{2}$, and divide both sides by $2\tau$ to get

$$\mathbb{E}_t[f(\mathbf{x}_{t+1}) + g(\mathbf{x}_{t+1}) - f(\mathbf{z}_t) - g(\mathbf{z}_t)] \leq \frac{1}{2\tau}\left(\|\mathbf{x}_t - \mathbf{z}_t\|^2 - \mathbb{E}_t\|\mathbf{x}_{t+1} - \mathbf{z}_t\|^2\right)$$
$$+ 2\tau L(1+\gamma)(h(\mathbf{x}_t) - h^*) + \tau\left(1 + \frac{1}{\gamma}\right)\sigma_*^2.$$

Rearranging terms, grouping $h(\mathbf{x}) = f(\mathbf{x}) + g(\mathbf{x})$ and using the notation $\mu = 2\tau L(1+\gamma)$, we get

$$\mathbb{E}_t[h(\mathbf{x}_{t+1}) - h(\mathbf{z}_t) - \mu h(\mathbf{x}_t) + \mu h^*] \leq \frac{1}{2\tau}\left(\|\mathbf{x}_t - \mathbf{z}_t\|^2 - \mathbb{E}_t\|\mathbf{x}_{t+1} - \mathbf{z}_t\|^2\right) + \tau\left(1 + \frac{1}{\gamma}\right)\sigma_*^2.$$

Taking total expectation gives us the required bound. □

### A.2. Reduction to last iterate bounds

The following result provides a last iterate bound for $\mathbb{E}[h(\mathbf{x}_T) - h^*]$. It follows the same arguments as in (Garrigos et al., 2025) and (Zamani & Glineur, 2025), but with different constants and terms due to the existence of a different left-hand side including $\mathbf{x}_{t+1}, \mathbf{x}_t, \mathbf{z}_t$.

**Lemma A.6** (Last-iterate reduction, see Lemma 3.3). *Suppose that for $t = 0, \ldots, T$ and every $\mathbf{z}_t \in \mathcal{F}(\mathbf{x}_0, \ldots, \mathbf{x}_t)$ it holds that*

$$\mathbb{E}[h(\mathbf{x}_{t+1}) - h(\mathbf{z}_t) - \mu h(\mathbf{x}_t) + \mu h^*] \leq \frac{1}{2\tau}\mathbb{E}[\|\mathbf{x}_t - \mathbf{z}_t\|^2 - \|\mathbf{x}_{t+1} - \mathbf{z}_t\|^2] + v, \tag{A.10}$$

*for some $0 \leq \mu < 1$ and $v > 0$.*

*Let $\mathbf{z}_t = (1 - p_t)\mathbf{x}_t + p_t\mathbf{z}_{t-1}$ for $t \geq 0$, with $p_0 = 1$, $\mathbf{z}_{-1} = \mathbf{x}^*$, and, for $t \geq 1$, we define*

$$p_t = \frac{\mu + T - t + 1}{T - t + 2}. \tag{A.11}$$

*Then, we have*

$$\mathbb{E}\left[h(\mathbf{x}_{T+1}) - h^*\right] \leq \frac{1}{\alpha_T}\left(h(\mathbf{x}_0) - h^* + \frac{1}{2\tau}\|\mathbf{x}_0 - \mathbf{x}^*\|^2\right) + \frac{v}{\alpha_T}\sum_{t=0}^{T}\alpha_t. \tag{A.12}$$

*where the sequence $(\alpha_t)$ is defined by $\alpha_{-1} = \alpha_0 = 1$ and*

$$\alpha_t = \frac{T - t + 2}{\mu + T - t + 1}\cdot\alpha_{t-1}, \qquad t = 1,\ldots,T.$$

*Proof.* Since $\mu < 1$, we have $p_t \in [0,1]$. Additionally, from the definition of $\alpha_t$, we have $\alpha_t p_t = \alpha_{t-1}$ for all $t \geq 0$. Consequently, $\alpha_t$ is non-decreasing. Multiplying both sides of (A.10) by $\alpha_t$, we have

$$\alpha_t\mathbb{E}[h(\mathbf{x}_{t+1}) - h(\mathbf{z}_t) - \mu h(\mathbf{x}_t) + \mu h^*] \leq \frac{\alpha_t}{2\tau}\mathbb{E}\|\mathbf{x}_t - \mathbf{z}_t\|^2 - \frac{\alpha_t}{2\tau}\mathbb{E}\|\mathbf{x}_{t+1} - \mathbf{z}_t\|^2 + \alpha_t v. \tag{A.13}$$

We estimate the first term on the right-hand side as

$$\alpha_t\|\mathbf{x}_t - \mathbf{z}_t\|^2 = \alpha_t\|p_t\mathbf{x}_t - p_t\mathbf{z}_{t-1}\|^2 = \alpha_t p_t^2\|\mathbf{x}_t - \mathbf{z}_{t-1}\|^2 \leq \alpha_t p_t\|\mathbf{x}_t - \mathbf{z}_{t-1}\|^2 = \alpha_{t-1}\|\mathbf{x}_t - \mathbf{z}_{t-1}\|^2,$$

where we used the definition of $\mathbf{z}_t$, $p_t \leq 1$ and the definition of $\alpha_t$ which gives $\alpha_t p_t = \alpha_{t-1}$.

Plugging this back into (A.13) gives

$$\alpha_t\mathbb{E}[h(\mathbf{x}_{t+1}) - h(\mathbf{z}_t) - \mu h(\mathbf{x}_t) + \mu h^*] \leq \frac{\alpha_{t-1}}{2\tau}\mathbb{E}\|\mathbf{x}_t - \mathbf{z}_{t-1}\|^2 - \frac{\alpha_t}{2\tau}\mathbb{E}\|\mathbf{x}_{t+1} - \mathbf{z}_t\|^2 + \alpha_t v. \tag{A.14}$$

Notice that we can telescope the right-hand side by summing from $t = 0$ to $T$. To complete this recursion, we define $\mathbf{z}_{-1} = \mathbf{x}^*$. Summing the inequality (A.14) from $t = 0$ to $T$ and telescoping give

$$\underbrace{\sum_{t=0}^{T}\alpha_t\mathbb{E}[h(\mathbf{x}_{t+1}) - h(\mathbf{z}_t) - \mu h(\mathbf{x}_t) + \mu h^*]}_{S} \leq \frac{\alpha_{-1}}{2\tau}\|\mathbf{x}_0 - \mathbf{z}_{-1}\|^2 + v\sum_{t=0}^{T}\alpha_t = \frac{1}{2\tau}\|\mathbf{x}_0 - \mathbf{x}^*\|^2 + v\sum_{t=0}^{T}\alpha_t, \tag{A.15}$$

where we used $\alpha_{-1} = 1$. We shall proceed to obtain a lower bound of the left-hand side of (A.15).

Unrolling the recursive definition for $\mathbf{z}_t$, we get:

$$\mathbf{z}_t = (1 - p_t)\mathbf{x}_t + p_t\mathbf{z}_{t-1} = \sum_{s=0}^{t}\left(\prod_{j=s+1}^{t}p_j\right)(1 - p_s)\mathbf{x}_s + \left(\prod_{j=0}^{t}p_j\right)\mathbf{x}^*, \tag{A.16}$$

where we used the convention $\prod_{j=t+1}^{t}p_j = 1$.

Using $p_j = \alpha_{j-1}/\alpha_j$, we have, for $0 \leq s \leq t$,

$$\prod_{j=s+1}^{t}p_j = \prod_{j=s+1}^{t}\frac{\alpha_{j-1}}{\alpha_j} = \frac{\alpha_s}{\alpha_t}, \qquad 1 - p_s = 1 - \frac{\alpha_{s-1}}{\alpha_s} = \frac{\alpha_s - \alpha_{s-1}}{\alpha_s}, \qquad \prod_{j=0}^{t}p_j = \prod_{j=0}^{t}\frac{\alpha_{j-1}}{\alpha_j} = \frac{\alpha_{-1}}{\alpha_t} = \frac{1}{\alpha_t}.$$

Therefore (A.16) becomes

$$\mathbf{z}_t = \sum_{s=0}^{t}\frac{\alpha_s}{\alpha_t}\cdot\frac{\alpha_s - \alpha_{s-1}}{\alpha_s}\mathbf{x}_s + \frac{1}{\alpha_t}\mathbf{x}^* = \sum_{s=0}^{t}\frac{\alpha_s - \alpha_{s-1}}{\alpha_t}\mathbf{x}_s + \frac{1}{\alpha_t}\mathbf{x}^*.$$

Notice that this is a convex combination of points $\mathbf{x}_0,\ldots,\mathbf{x}_t,\mathbf{x}^*$ as the weights sum to 1, that is,

$$\frac{1}{\alpha_t} + \sum_{s=0}^{t}\frac{\alpha_s - \alpha_{s-1}}{\alpha_t} = \frac{1}{\alpha_t}(1 + \alpha_t - \alpha_{-1}) = 1,$$

since $\alpha_{-1} = 1$ and each weight is between $[0, 1]$ as $\alpha_t$ is non-decreasing. So this definition of $\mathbf{z}_t$ agrees with our earlier assumption that $\mathbf{z}_t \in \mathcal{F}(\mathbf{x}_0, \ldots, \mathbf{x}_t, \mathbf{x}^*)$. As $h$ is convex, by Jensen's inequality we have,

$$h(\mathbf{z}_t) \leq \frac{1}{\alpha_t} \left( h(\mathbf{x}^*) + \sum_{s=0}^{t} (\alpha_s - \alpha_{s-1}) h(\mathbf{x}_s) \right). \tag{A.17}$$

Substituting (A.17) into the left-hand side of the inequality (A.15)

$$
\begin{aligned}
S &\geq \sum_{t=0}^{T} \alpha_t \mathbb{E} \left[ h(\mathbf{x}_{t+1}) - \frac{1}{\alpha_t} \left( h^* + \sum_{s=0}^{t} (\alpha_s - \alpha_{s-1}) h(\mathbf{x}_s) \right) - \mu h(\mathbf{x}_t) + \mu h^* \right] \\
&= \sum_{t=0}^{T} \mathbb{E} \left[ \alpha_t h(\mathbf{x}_{t+1}) - \sum_{s=0}^{t} (\alpha_s - \alpha_{s-1}) h(\mathbf{x}_s) - \mu \alpha_t h(\mathbf{x}_t) + (\mu \alpha_t - 1) h^* \right].
\end{aligned}
\tag{A.18}
$$

We next estimate the second term inside the bracket.

$$
\begin{aligned}
\sum_{s=0}^{t} (\alpha_s - \alpha_{s-1}) \left( h(\mathbf{x}_s) - h^* + h^* \right) &= \sum_{s=0}^{t} (\alpha_s - \alpha_{s-1}) \left( h(\mathbf{x}_s) - h^* \right) + h^* (\alpha_t - \alpha_{-1}) \\
&= \sum_{s=0}^{t} (\alpha_s - \alpha_{s-1}) \left( h(\mathbf{x}_s) - h^* \right) + \alpha_t h^* - h^*,
\end{aligned}
$$

using $\alpha_{-1} = 1$. Combining this with (A.18), we obtain a lower bound on the left-hand side of (A.15):

$$S \geq \sum_{t=0}^{T} \mathbb{E} \left[ \alpha_t (h(\mathbf{x}_{t+1}) - h^*) - \mu \alpha_t (h(\mathbf{x}_t) - h^*) - \sum_{s=0}^{t} (\alpha_s - \alpha_{s-1}) \left[ h(\mathbf{x}_s) - h^* \right] \right]. \tag{A.19}$$

We next proceed to show that this lower bound gives us an expression for the last iterate. Define

$$r_t = h(\mathbf{x}_t) - h^*, \tag{A.20}$$

then (A.19) becomes

$$S \geq \mathbb{E} \left[ \sum_{t=0}^{T} \alpha_t r_{t+1} - \mu \sum_{t=0}^{T} \alpha_t r_t - \sum_{t=0}^{T} \sum_{s=0}^{t} (\alpha_s - \alpha_{s-1}) r_s \right]. \tag{A.21}$$

For the double sum, we change the order of summation

$$\sum_{t=0}^{T} \sum_{s=0}^{t} (\alpha_s - \alpha_{s-1}) r_s = \sum_{s=0}^{T} \sum_{t=s}^{T} (\alpha_s - \alpha_{s-1}) r_s = \sum_{s=0}^{T} (\alpha_s - \alpha_{s-1})(T - s + 1) r_s.$$

Using this in (A.21) gives

$$S \geq \mathbb{E} \left[ \sum_{t=0}^{T} \alpha_t r_{t+1} - \mu \sum_{t=0}^{T} \alpha_t r_t - \sum_{t=0}^{T} (\alpha_t - \alpha_{t-1})(T - t + 1) r_t \right]. \tag{A.22}$$

We can rewrite the first term in the bracket as

$$\sum_{t=0}^{T} \alpha_t r_{t+1} = \alpha_T r_{T+1} + \sum_{t=0}^{T-1} \alpha_t r_{t+1} = \alpha_T r_{T+1} + \sum_{t=1}^{T} \alpha_{t-1} r_t.$$

The second term inside the bracket in (A.22) can be rewritten as

$$\mu \sum_{t=0}^{T} \alpha_t r_t = \mu \alpha_0 r_0 + \mu \sum_{t=1}^{T} \alpha_t r_t.$$

The third term inside the bracket in (A.22) can be rewritten as

$$\sum_{t=0}^{T}(\alpha_t - \alpha_{t-1})(T - t + 1)r_t = (\alpha_0 - 1)(T+1)r_0 + \sum_{t=1}^{T}(\alpha_t - \alpha_{t-1})(T - t + 1)r_t.$$

where in the last line we used $\alpha_{-1} = 1$.

Combining the last three display equations in (A.22) gives

$$S \geq \mathbb{E}\left[\alpha_T r_{T+1} + \sum_{t=1}^{T}\alpha_{t-1}r_t - \mu\alpha_0 r_0 - \mu\sum_{t=1}^{T}\alpha_t r_t - (\alpha_0 - 1)(T+1)r_0 - \sum_{t=1}^{T}(\alpha_t - \alpha_{t-1})(T - t + 1)r_t\right]. \quad (A.23)$$

We now group the terms with $r_0$ and $r_t$. For terms with $r_0$, we have

$$-\mu\alpha_0 r_0 - (\alpha_0 - 1)(T+1)r_0 = -r_0\left(\mu\alpha_0 + (\alpha_0 - 1)(T+1)\right).$$

For terms with $r_t$ in (A.23), we have

$$\sum_{t=1}^{T} r_t\left(\alpha_{t-1} - \mu\alpha_t - (\alpha_t - \alpha_{t-1})(T - t + 1)\right) = \sum_{t=1}^{T} r_t\left(\alpha_{t-1}(T - t + 2) - \alpha_t(\mu + T - t + 1)\right).$$

Plugging these into (A.23) gives

$$S \geq \mathbb{E}\left[\alpha_T r_{T+1} + \sum_{t=1}^{T} r_t\left(\alpha_{t-1}(T - t + 2) - \alpha_t(\mu + T - t + 1)\right) - r_0\left(\mu\alpha_0 + (\alpha_0 - 1)(T+1)\right)\right]. \quad (A.24)$$

Now notice that from our definition of $\alpha_t$, we have that for all $t \geq 1$,

$$\alpha_{t-1}(T - t + 2) - \alpha_t(\mu + T - t + 1) = 0.$$

Using this in (A.24) gives

$$S \geq \mathbb{E}\left[\alpha_T r_{T+1} - r_0\left(\mu\alpha_0 + (\alpha_0 - 1)(T+1)\right)\right] = \mathbb{E}\left[\alpha_T r_{T+1} - r_0\mu\right], \quad (A.25)$$

since $\alpha_0 = 1$ which gives $\mu\alpha_0 + (\alpha_0 - 1)(T+1) = \mu$. Combining (A.15) and (A.25), we have the inequality

$$\mathbb{E}\left[\alpha_T r_{T+1}\right] \leq \frac{1}{2\tau}\|\mathbf{x}_0 - \mathbf{x}^*\|^2 + v\sum_{t=0}^{T}\alpha_t + r_0\mu.$$

Dividing by $\alpha_T$, using $\mu < 1$, and the definition of $r_t$ from (A.20), we obtain the assertion. $\qquad\square$

### A.3. Technical lemmas needed for Theorem A.1

Now we proceed to obtain a concrete bound for polynomial step sizes where $\tau = \mathcal{O}(\frac{1}{\sqrt{T}})$. Before we do that, we first prove some technical lemmas that bound $\alpha_T$, $\sum_{t=0}^{T}\alpha_t$, and $T^\mu$. We use estimations similar to Garrigos et al. (2025).

**Lemma A.7** (Bounds on the $\alpha_t$ sequence). *Let $T \geq 1$ and fix $\mu \in (0, 1)$. Define $(\alpha_t)_{t=0}^{T}$ recursively by*

$$\alpha_0 = \alpha_{-1} = 1, \qquad \alpha_t = \frac{T - t + 2}{T - t + 1 + \mu}\alpha_{t-1}, \quad t = 1, \ldots, T.$$

*Then it holds that*

1. *$\alpha_T \geq \frac{(T+1)^{1-\mu}}{2^{1-\mu}}$,*

2. *$\frac{\sum_{t=1}^{T}\alpha_t}{\alpha_T} \leq 8T^\mu \ln(T+1)$,*

*Proof.* By unrolling the recursion of $\alpha_t$, we have

$$\alpha_t = \prod_{k=1}^{t} \frac{T-k+2}{T-k+1+\mu}.$$

Recall the gamma function, $\Gamma(n) = (n-1)!$. So we can write the numerator of $\alpha_t$ using the gamma function as,

$$\prod_{k=1}^{t} (T-k+2) = \frac{\Gamma(T+2)}{\Gamma(T-t+2)}.$$

And similarly, we can rewrite the denominator with the gamma function as

$$\prod_{k=1}^{t} (T-k+1+\mu) = \frac{\Gamma(T+\mu+1)}{\Gamma(T-t+\mu+1)}.$$

Combining the last two estimates, we can rewrite $\alpha_t$ as

$$\alpha_t = \frac{\Gamma(T+1+1)\Gamma(T-t+1+\mu)}{\Gamma(T+1+\mu)\Gamma(T-t+1+1)}.$$

By Gautschi's inequality (Gautschi, 1959), we have that for all $\mathbf{x} > 0$ and all $c \in [0, 1]$,

$$\mathbf{x}^{1-c} \leq \frac{\Gamma(\mathbf{x}+1)}{\Gamma(\mathbf{x}+c)} \leq (\mathbf{x}+1)^{1-c}.$$

Applying this for $\mathbf{x} = T+1$ and $\mathbf{x} = T-t+1$,

$$(T+1)^{1-\mu} \leq \frac{\Gamma(T+1+1)}{\Gamma(T+1+\mu)} \leq (T+2)^{1-\mu},$$

and similarly

$$(T-t+1)^{1-\mu} \leq \frac{\Gamma(T-t+1+1)}{\Gamma(T-t+1+\mu)} \leq (T-t+2)^{1-\mu}.$$

So we have the lower bound for $\alpha_t$:

$$\alpha_t \geq \frac{(T+1)^{1-\mu}}{(T-t+2)^{1-\mu}}.$$

In particular, for $t = T$, we have

$$\alpha_T \geq \frac{(T+1)^{1-\mu}}{2^{1-\mu}}. \tag{A.26}$$

We also have an upper bound for $\alpha_t$ as

$$\alpha_t \leq \frac{(T+2)^{1-\mu}}{(T-t+1)^{1-\mu}}.$$

Now, we sum this inequality over $t$ and upper bound the sum with the integral test,

$$\sum_{t=1}^{T} \alpha_t \leq \sum_{t=1}^{T} \frac{(T+2)^{1-\mu}}{(T-t+1)^{1-\mu}} \leq (T+2)^{1-\mu} \sum_{t=1}^{T} \frac{1}{(T-t+1)^{1-\mu}} \leq (T+2)^{1-\mu} \left(1 + \frac{T^{\mu}-1}{\mu}\right).$$

We use this bound and (A.26) to get

$$\frac{\sum_{t=1}^{T} \alpha_t}{\alpha_T} \leq (T+2)^{1-\mu} \left(1 + \frac{T^{\mu}-1}{\mu}\right) \cdot \frac{2^{1-\mu}}{(T+1)^{1-\mu}}$$

$$= \left(\frac{2(T+2)}{T+1}\right)^{1-\mu} \left(1 + \frac{T^{\mu}-1}{\mu}\right)$$

$$\leq 4\left(1 + \frac{T^{\mu}-1}{\mu}\right),$$

where in the last line we used the fact that $\frac{T+2}{T+1} \leq 2$ and $\mu < 1$. So we have the estimate for $\frac{\sum_{t=1}^{T} \alpha_t}{\alpha_T}$:

$$\frac{\sum_{t=1}^{T} \alpha_t}{\alpha_T} \leq 4 \left(1 + \frac{T^\mu - 1}{\mu}\right).$$

Finally, we bound the right-hand side. Define

$$\psi(T) = 8T^\mu \ln (T+1) - 4 \left(1 + \frac{T^\mu - 1}{\mu}\right).$$

We will show that $\psi(T) \geq 0$ for every $T \geq 1$. We compute the derivative of $\psi$:

$$\psi'(T) = 8\mu T^{\mu-1} \ln (T+1) + \frac{8T^\mu}{T+1} - 4T^{\mu-1} = 4T^{\mu-1}\left(2\mu \ln (T+1) + \frac{2T}{T+1} - 1\right).$$

Since $T \geq 1$, we have $\ln (T+1) \geq 0$ and $\frac{T-1}{T+1} \geq 0$, and hence,

$$2a \ln (T+1) + \frac{2T}{T+1} - 1 = 2\mu \ln (T+1) + \frac{T-1}{T+1} \geq 0,$$

and thus $\psi'(T) \geq 0$ for every $T \geq 1$. Furthermore, $\psi(1) = 8\ln(2) - 4 > 0$. Hence, $\psi$ is nondecreasing and $\psi(1) > 0$, this implies $\psi(T) \geq 0$ for every $T \geq 1$. This gives us the assertion. $\qquad \square$

We now show that $T^\mu$ is a constant given that $\tau$ has a polynomial dependence on $T$ and for the specific choice of $\mu = 2\tau L(1+\gamma)$ we had obtained Lemma 3.2.

**Lemma A.8** ($T^\mu$ for polynomially decaying step sizes)**.** *Let $T \geq 1$ and $C > 2$. If we have $\tau = \frac{1}{CL\sqrt{T}}$ and $\mu = 2\tau L(1+\gamma)$, where $\gamma = \frac{1-2\tau L}{1+2\tau L}$, then it follows that $\tau L = \frac{1}{C\sqrt{T}} \leq \frac{1}{C} < \frac{1}{2}$, and*

$$T^\mu \leq \exp\left(\frac{8}{eC}\right).$$

*Proof.* We use the definition of $\mu$ to derive

$$T^\mu = T^{2\tau L(1+\gamma)} = \exp\left(2\tau L(1+\gamma)\ln T\right) = \exp\left(\frac{2(1+\gamma)\ln T}{C\sqrt{T}}\right). \tag{A.27}$$

We now use the fact that $\frac{\ln T}{\sqrt{T}} \leq \frac{2}{e}$ for every $T \geq 1$ in (A.27) to get

$$T^\mu \leq \exp\left(\frac{4(1+\gamma)}{eC}\right).$$

Using our definition of $\gamma = \frac{1-2\tau L}{1+2\tau L}$, we can estimate $1 + \gamma = \frac{2}{1+2\tau L} \leq 2$, since $1 + 2\tau L \geq 1$. Substituting this back gives the last assertion. $\qquad \square$

## A.4. Proof of Theorem A.1: Last-iterate analysis

We combine the above results to obtain the last iterate bound for step size $\tau = \frac{1}{CL\sqrt{T}}$ for $C > 2$.

*Proof of Theorem A.1.* From Lemma 3.2 we obtained the following

$$\mathbb{E}\left[h(\mathbf{x}_{t+1}) - h(\mathbf{z}_t) - \mu h(\mathbf{x}_t) + \mu h^*\right] \leq \frac{1}{2\tau}\mathbb{E}\left[\|\mathbf{x}_t - \mathbf{z}_t\|^2 - \|\mathbf{x}_{t+1} - \mathbf{z}_t\|^2\right] + v,$$

where $\mu = 2\tau L(1+\gamma)$ and $v = (1 + 1/\gamma)\sigma_*^2\tau$ for any $\gamma > 0$. In particular, we choose $\gamma = \frac{1-2\tau L}{1+2\tau L}$. Then, notice that $\mu < 1$ since $\tau L < \frac{1}{2}$. Now, we can apply Lemma A.6 to get the following bound on the last iterate:

$$\mathbb{E}\left[h(\mathbf{x}_{T+1}) - h^*\right] \leq \frac{1}{2\tau\alpha_T}\|\mathbf{x}_0 - \mathbf{x}^*\|^2 + \frac{1}{\alpha_T}(h(\mathbf{x}_0) - h^*) + +\frac{v\alpha_0}{\alpha_T} + \frac{v}{\alpha_T}\sum_{t=1}^{T}\alpha_t. \tag{A.28}$$

Now, Lemma A.7 tells us that

$$\alpha_T \geq \frac{(T+1)^{1-\mu}}{2^{1-\mu}} \geq \frac{T^{1-\mu}}{2} \quad \text{and} \quad \frac{\sum_{t=1}^{T} \alpha_t}{\alpha_T} \leq 8T^{\mu} \ln(T+1).$$

Plugging these into (A.28), we get

$$\mathbb{E}\left[h(\mathbf{x}_{T+1}) - h^*\right] \leq T^{\mu} \left[ \frac{\|\mathbf{x}_0 - \mathbf{x}^*\|^2}{\tau T} + \frac{2}{T}\left(h(\mathbf{x}_0) - h^* + v\right) + 8v \ln(T+1) \right] \tag{A.29}$$

From our step size, $\tau = \frac{1}{CL\sqrt{T}}$ we get $\tau L \leq \frac{1}{C}$ and hence, $\frac{1}{1-2\tau L} \leq \frac{C}{C-2}$. From our choice of $\gamma = \frac{1-2\tau L}{1+2\tau L}$, we get

$$v = (1+1/\gamma)\sigma_*^2 \tau = \frac{2}{1-2\tau L}\sigma_*^2 \tau \leq \frac{2\sigma_*^2 C\tau}{C-2} = \frac{2\sigma_*^2}{(C-2)L\sqrt{T}}.$$

Substituting our step size $\tau$, the above bound and the bound for $T^{\mu}$ from Lemma A.8 into (A.29) gives the assertion. □

# B. Main results for Section 4

We now provide the main results for the (RIPM) algorithm, which are presented with a constant step size depending on the horizon $T$. For the statement in the main text, that is, Theorem 4.1, we used $C = 5$ and simplified the bounds.

**Theorem B.1** (Last-iterate convergence, incremental proximal, generalized version of Theorem 4.1). *Consider the (RIPM) algorithm with step size $\tau = \frac{1}{CL\sqrt{T}}$ for some $C > 4$. Let Assumptions 2.1 and 2.2 hold. Then, the last iterate $\mathbf{x}_{T+1}$ satisfies*

$$\mathbb{E}[h(\mathbf{x}_{T+1}) - h^*] \leq \exp\left(\frac{8}{eC}\right) \left[ \frac{CL\|\mathbf{x}_0 - \mathbf{x}^*\|^2}{\sqrt{T}} + \frac{2}{T}(h(\mathbf{x}_0) - h^*) + \frac{4\sigma_*^2}{(C-2)LT^{1.5}} + \frac{16\sigma_*^2 \ln(T+1)}{(C-2)L\sqrt{T}} \right.$$
$$\left. + \frac{16m^2 L_g^2}{CLT^{1.5}} + \frac{64m^2 L_g^2 \ln(T+1)}{CL\sqrt{T}} \right].$$

## B.1. One iteration lemma for (RIPM)

We first prove the following lemma that is the corresponding result to Lemma 3.2.

**Lemma B.2** (Per-iteration descent, incremental proximal, see Lemma 4.2). *Let Assumptions 2.1 and 2.2 hold; and let $(\mathbf{x}_t)_{t=0}^{T}$ be generated by (RIPM) with a $\tau$ satisfying $0 < \tau L < \frac{1}{4}$. Then for all $t = 0, \dots T$ and $\mathbf{z}_t$ in $\mathcal{F}(\mathbf{x}_0, ..., \mathbf{x}_t)$ we have*

$$\mathbb{E}\left[h(\mathbf{x}_{t+1}) - h(\mathbf{z}_t) - \mu h(\mathbf{x}_t) + \mu h^*\right] \leq \frac{1}{2\tau}\mathbb{E}\left[\|\mathbf{x}_t - \mathbf{z}_t\|^2 - \|\mathbf{x}_{t+1} - \mathbf{z}_t\|^2\right] + \left(1 + \frac{1}{\gamma}\right)\sigma_*^2 \tau + 8\tau m^2 L_g^2, \tag{B.1}$$

*where $\mu = 2\tau L(1+\gamma)$ for some $\gamma > 0$.*

*Proof.* We can apply Lemma A.4 here by using $mg_i$ instead of $g$ as defined in our proximal step. We obtain,

$$\|\mathbf{x}_{t+1} - \mathbf{z}_t\|^2 - \|\mathbf{x}_t - \mathbf{z}_t\|^2 \leq 2\tau(f_{j_t}(\mathbf{z}_t) - f_{j_t}(\mathbf{x}_t) + f(\mathbf{x}_t) - f(\mathbf{x}_{t+1})) + 2\tau m(g_{i_t}(\mathbf{z}_t) - g_{i_t}(\mathbf{x}_{t+1}))$$
$$+ \frac{\tau}{\delta_1}\|\nabla f_{j_t}(\mathbf{x}_t) - \nabla f(\mathbf{x}_t)\|^2 + (\tau\delta_1 + \tau L - 1)\|\mathbf{x}_t - \mathbf{x}_{t+1}\|^2. \tag{B.2}$$

Here, taking conditional expectation, and using $\mathbb{E}\|X - \mathbb{E}X\|^2 \leq \mathbb{E}\|X\|^2$, we have

$$\mathbb{E}_t\|\mathbf{x}_{t+1} - \mathbf{z}_t\|^2 - \|\mathbf{x}_t - \mathbf{z}_t\|^2 \leq 2\tau\mathbb{E}_t[f(\mathbf{z}_t) - f(\mathbf{x}_{t+1})] + 2\tau m\mathbb{E}_t(g_{i_t}(\mathbf{z}_t) - g_{i_t}(\mathbf{x}_{t+1}))$$
$$+ \frac{\tau}{\delta_1}\mathbb{E}_t\|\nabla f_{j_t}(\mathbf{x}_t)\|^2 + (\tau\delta_1 + \tau L - 1)\mathbb{E}_t\|\mathbf{x}_t - \mathbf{x}_{t+1}\|^2. \tag{B.3}$$

Consider the term $2\tau m(g_{i_t}(\mathbf{z}_t) - g_{i_t}(\mathbf{x}_{t+1}))$. We can rewrite this as:

$$2\tau m(g_{i_t}(\mathbf{z}_t) - g_{i_t}(\mathbf{x}_{t+1})) = 2\tau m(g_{i_t}(\mathbf{z}_t) - g_{i_t}(\mathbf{x}_t)) + 2\tau m(g_{i_t}(\mathbf{x}_t) - g_{i_t}(\mathbf{x}_{t+1}))$$
$$\leq 2\tau m\left(g_{i_t}(\mathbf{z}_t) - \frac{1}{m}g(\mathbf{x}_{t+1})\right) + 2\tau m\left(\frac{1}{m}g(\mathbf{x}_{t+1}) - g_{i_t}(\mathbf{x}_t)\right) + 2\tau m L_g\|\mathbf{x}_t - \mathbf{x}_{t+1}\|,$$

where we used the Lipschitzness of $g_i$. Taking conditional expectation of the above (using $\mathbb{E}_t[g_i(\mathbf{u})] = \frac{1}{m}g(\mathbf{u})$ where $\mathbf{u}$ is measurable with respect to the conditioning of the expectation) and then total expectation, we have

$$2\tau m\mathbb{E}[g_{i_t}(\mathbf{z}_t) - g_{i_t}(\mathbf{x}_{t+1})] \leq 2\tau\mathbb{E}[g(\mathbf{z}_t) - g(\mathbf{x}_{t+1})] + 2\tau\mathbb{E}[g(\mathbf{x}_{t+1}) - g(\mathbf{x}_t)] + 2\tau m L_g \mathbb{E}\|\mathbf{x}_t - \mathbf{x}_{t+1}\|. \tag{B.4}$$

As each $g_i$ is Lipschitz, notice that $g$ is also Lipschitz with the constant $mL_g$ since by triangle inequality, we have

$$|g(\mathbf{x}) - g(\mathbf{y})| = \left|\sum_{i=1}^m g_i(\mathbf{x}) - \sum_{i=1}^m g_i(\mathbf{y})\right| \leq \sum_{i=1}^m |g_i(\mathbf{x}) - g_i(\mathbf{y})| \leq mL_g\|\mathbf{x} - \mathbf{y}\|.$$

Then, using the Lipschitzness of $g$ in (B.4), we have

$$2\tau m\mathbb{E}[g_{i_t}(\mathbf{z}_t) - g_{i_t}(\mathbf{x}_{t+1})] \leq 2\tau\mathbb{E}[g(\mathbf{z}_t) - g(\mathbf{x}_{t+1})] + 4\tau m L_g \mathbb{E}\|\mathbf{x}_t - \mathbf{x}_{t+1}\|. \tag{B.5}$$

Next, for (B.3), we can use Lemma A.2 to bound the third term on the right-hand side, take total expectation and use (B.5) to bound the second term on the right-hand side to deduce

$$\mathbb{E}[\|\mathbf{x}_{t+1} - \mathbf{z}_t\|^2 - \|\mathbf{x}_t - \mathbf{z}_t\|^2] \leq 2\tau\mathbb{E}[f(\mathbf{z}_t) - f(\mathbf{x}_{t+1}) + g(\mathbf{z}_t) - g(\mathbf{x}_{t+1})] + \frac{2\tau L(1+\gamma)}{\delta_1}\mathbb{E}[h(\mathbf{x}_t) - h^*]$$

$$+ (\tau\delta_1 + \tau L - 1)\mathbb{E}\|\mathbf{x}_t - \mathbf{x}_{t+1}\|^2 + 4\tau m L_g\mathbb{E}\|\mathbf{x}_t - \mathbf{x}_{t+1}\| + \frac{\tau(\gamma+1)}{\gamma\delta_1}\sigma_*^2. \tag{B.6}$$

We define $\delta_1 = \frac{3}{4\tau} - L > 0$, which gives $1 - \tau\delta_1 - \tau L > 0$. Hence, we estimate the second from last term here by using Young's inequality to get

$$4\tau m L_g\|\mathbf{x}_t - \mathbf{x}_{t+1}\| \leq (1 - \tau\delta_1 - \tau L)\|\mathbf{x}_t - \mathbf{x}_{t+1}\|^2 + \frac{4\tau^2 m^2 L_g^2}{1 - \tau\delta_1 - \tau L}.$$

Applying this estimate in (B.6) and rearranging, we have

$$2\tau\mathbb{E}[h(\mathbf{x}_{t+1}) - h(\mathbf{z}_t)] \leq \mathbb{E}[\|\mathbf{x}_t - \mathbf{z}_t\|^2 - \|\mathbf{x}_{t+1} - \mathbf{z}_t\|^2] + \frac{2\tau L(1+\gamma)}{\delta_1}\mathbb{E}[h(\mathbf{x}_t) - h^*] + \frac{4\tau^2 m^2 L_g^2}{1 - \tau\delta_1 - \tau L} + \frac{\tau(\gamma+1)}{\gamma\delta_1}\sigma_*^2.$$

We next use $\frac{\tau}{\delta_1} = \frac{4\tau^2}{3 - 4\tau L} \leq 2\tau^2$, which uses $\tau L < \frac{1}{4}$ and $3 - 4\tau L > 2$, divide both sides by $2\tau$ and rearrange to obtain

$$\mathbb{E}[h(\mathbf{x}_{t+1}) - h(\mathbf{z}_t) - 2\tau L(1+\gamma)h(\mathbf{x}_t) + 2\tau L(1+\gamma)h^*] \leq \frac{1}{2\tau}\mathbb{E}[\|\mathbf{x}_t - \mathbf{z}_t\|^2 - \|\mathbf{x}_{t+1} - \mathbf{z}_t\|^2]$$

$$+ \left(1 + \frac{1}{\gamma}\right)\tau\sigma_*^2 + \frac{2\tau m^2 L_g^2}{1 - \tau\delta_1 - \tau L}.$$

Finally, we utilize the equality $1 - \tau\delta_1 - \tau L = \frac{1}{4}$ for the last term and then use the notation $\mu = 2\tau L(1+\gamma)$. $\qquad\square$

## B.2. Proof of Theorem B.1: Last-iterate analysis of (RIPM)

*Proof.* From (B.7), we obtained

$$\mathbb{E}[h(\mathbf{x}_{t+1}) - h(\mathbf{z}_t) - \mu h(\mathbf{x}_t) + \mu h^*] \leq \frac{1}{2\tau}\mathbb{E}[\|\mathbf{x}_t - \mathbf{z}_t\|^2 - \|\mathbf{x}_{t+1} - \mathbf{z}_t\|^2] + v, \tag{B.7}$$

where $\mu = 2\tau L(1+\gamma)$ and $v = (1 + 1/\gamma)\sigma_*^2\tau + 8\tau m^2 L_g^2$ for some $\gamma > 0$. In particular, we choose $\gamma = \frac{1-2\tau L}{1+2\tau L}$, so that $\mu = \frac{4\tau L}{1+2\tau L} < 1$ since $\tau < \frac{1}{4L}$. Hence, we can apply Lemma A.6 to obtain the following bound for the last iterate:

$$\mathbb{E}[h(\mathbf{x}_{T+1}) - h^*] \leq \frac{1}{2\tau\alpha_T}\|\mathbf{x}_0 - \mathbf{x}^*\|^2 + \frac{v\alpha_0}{\alpha_T} + \frac{v}{\alpha_T}\sum_{t=1}^T \alpha_t + \frac{1}{\alpha_T}[h(\mathbf{x}_0) - h^*].$$

Expanding $\mu$ and $v$, we obtain

$$\mathbb{E}[h(\mathbf{x}_{T+1}) - h^*] \leq \underbrace{\frac{1}{2\tau\alpha_T}\|\mathbf{x}_0 - \mathbf{x}^*\|^2 + \frac{v\alpha_0}{\alpha_T} + \frac{v}{\alpha_T}\sum_{t=1}^T \alpha_t + \frac{1}{\alpha_T}\mathbb{E}[h(\mathbf{x}_0) - h^*]}_{1} + \underbrace{\frac{8\tau m^2 L_g^2}{\alpha_T}\sum_{t=0}^T \alpha_t}_{2}. \tag{B.8}$$

It suffices to bound (2) in (B.8) since we already have bounds for (1) from Theorem A.1. Splitting this term, we have

$$\frac{8\tau m^2 L_g^2}{\alpha_T} \sum_{t=0}^{T} \alpha_t = 8\tau m^2 L_g^2 \frac{\alpha_0}{\alpha_T} + 8\tau m^2 L_g^2 \frac{\sum_{t=1}^{T} \alpha_t}{\alpha_T}. \tag{B.9}$$

Consider the first term on the right-hand side. Using $\alpha_0 = 1$, Lemma A.7, and the definition of $\tau$, we have

$$8\tau m^2 L_g^2 \frac{\alpha_0}{\alpha_T} \le 8\tau m^2 L_g^2 \cdot \frac{2}{T^{1-\mu}} = T^{\mu} \frac{16 m^2 L_g^2}{CLT^{1.5}}.$$

For the second term on the right-hand side of (B.9), using Lemma A.7 and the definition of $\tau$ gives

$$8\tau m^2 L_g^2 \frac{\sum_{t=1}^{T} \alpha_t}{\alpha_T} \le T^{\mu} \frac{64 m^2 L_g^2 \ln(T+1)}{CL\sqrt{T}}.$$

Using Lemma A.8, and combining the above two bounds, we have

$$\frac{8\tau m^2 L_g^2}{\alpha_T} \sum_{t=0}^{T} \alpha_t \le \exp\left(\frac{8}{eC}\right) \left(\frac{16 m^2 L_g^2}{CLT^{1.5}} + \frac{64 m^2 L_g^2 \ln(T+1)}{CL\sqrt{T}}\right), \tag{B.10}$$

where we used the upper bound for $T^{\mu}$ from Lemma A.8. Now plugging in the estimates for (1) from Theorem A.1 equations (A.28) and (A.29) and the estimates for (2) from (B.10) into (B.8) gives the assertion. $\square$

## B.3. Proof of Section 4.3

Before proving Corollary 4.3, we would like to clarify that the proof can be simplified by staying in $x_t$ instead of moving to $x_{t+1}$. Additionally, the bound obtained below can be improved in terms of the non dominant terms by removing the $g(x_0) - g(x^*)$ term from the right-hand side as $\mu = 0$. However, our goal was to reuse the previous lemma to avoid repeating several calculations. A cleaner proof can be obtained by redoing the calculations for the case $\mu = 0$ and staying in $x_t$ instead of $x_{t+1}$.

*Proof of Corollary 4.3.* We begin from (A.5)

$$\|\mathbf{x}_{t+1} - \mathbf{z}_t\|^2 - \|\mathbf{x}_t - \mathbf{z}_t\|^2 = \|\mathbf{x}_{t+1} - \mathbf{x}_t\|^2 + 2\langle \mathbf{x}_{t+1} - \mathbf{x}_t, \mathbf{x}_t - \mathbf{z}_t \rangle$$

Similar to (A.6) in the proof of Lemma A.4 we use the optimality condition of the proximal point step, to obtain

$$\|\mathbf{x}_{t+1} - \mathbf{z}_t\|^2 - \|\mathbf{x}_t - \mathbf{z}_t\|^2 \le -\|\mathbf{x}_{t+1} - \mathbf{x}_t\|^2 + 2\tau(g_{j_t}(\mathbf{z}_t) - g_{j_t}(\mathbf{x}_{t+1})). \tag{B.11}$$

Now, we estimate the term $2\tau[g_{j_t}(\mathbf{z}_t) - g_{j_t}(\mathbf{x}_{t+1})]$ as follows:

$$2\tau(g_{j_t}(\mathbf{z}_t) - g_{j_t}(\mathbf{x}_{t+1})) = 2\tau[g_{j_t}(\mathbf{z}_t) - g(\mathbf{x}_{t+1})] + 2\tau[g(\mathbf{x}_{t+1}) - g_{j_t}(\mathbf{x}_t)] + 2\tau[g_{j_t}(\mathbf{x}_t) - g_{j_t}(\mathbf{x}_{t+1})]$$

We take conditional expectation, and then total expectation, and use Lipschitzness of $g_j$ to get

$$2\tau\mathbb{E}[g_{j_t}(\mathbf{z}_t) - g_{j_t}(\mathbf{x}_{t+1})] \le 2\tau\mathbb{E}[g(\mathbf{z}_t) - g(\mathbf{x}_{t+1})] + 2\tau\mathbb{E}[g(\mathbf{x}_{t+1}) - g(\mathbf{x}_t)] + 2\tau L_g \mathbb{E}\|\mathbf{x}_t - \mathbf{x}_{t+1}\|$$

$$\le 2\tau\mathbb{E}[g(\mathbf{z}_t) - g(\mathbf{x}_{t+1})] + 4\tau L_g \mathbb{E}\|\mathbf{x}_t - \mathbf{x}_{t+1}\|,$$

where in the last line we used the Lipschitzness of $g$ to estimate $\mathbb{E}[g(\mathbf{x}_{t+1}) - g(\mathbf{x}_t)]$. Plugging this estimate back into (B.11) after taking total expectation, we have

$$\mathbb{E}[\|\mathbf{x}_{t+1} - \mathbf{z}_t\|^2 - \|\mathbf{x}_t - \mathbf{z}_t\|^2] \le 2\tau\mathbb{E}[g(\mathbf{z}_t) - g(\mathbf{x}_{t+1})] + 4\tau L_g \mathbb{E}\|\mathbf{x}_t - \mathbf{x}_{t+1}\| - \mathbb{E}\|\mathbf{x}_{t+1} - \mathbf{x}_t\|^2.$$

Using Young's inequality, we can estimate the second term on the right-hand side here as

$$4\tau L_g \mathbb{E}\|\mathbf{x}_t - \mathbf{x}_{t+1}\| \le 4\tau^2 L_g^2 + \mathbb{E}\|\mathbf{x}_t - \mathbf{x}_{t+1}\|^2.$$

Substituting this into the previous inequality, rearrange and divide by $2\tau$ to deduce

$$\mathbb{E}[g(\mathbf{x}_{t+1}) - g(\mathbf{z}_t)] \leq \frac{1}{2\tau}\mathbb{E}[\|\mathbf{x}_t - \mathbf{z}_t\|^2 - \|\mathbf{x}_{t+1} - \mathbf{z}_t\|^2] + 2\tau L_g^2.$$

Now, we can apply Lemma A.6 with $\mu = 0$ and $v = 2\tau L_g^2$ to obtain the following bound for the last iterate:

$$\mathbb{E}[g(\mathbf{x}_{T+1}) - g^*] \leq \frac{1}{\alpha_T}\left(g(\mathbf{x}_0) - g^* + \frac{1}{2\tau}\|\mathbf{x}_0 - \mathbf{x}^*\|^2\right) + \frac{v}{\alpha_T}\sum_{t=0}^{T}\alpha_t, \tag{B.12}$$

where

$$\alpha_t = \frac{T - t + 2}{T - t + 1}\alpha_{t-1},$$

with $\alpha_0 = \alpha_{-1} = 1$. Now, unrolling the recursion for $\alpha_t$ gives

$$\alpha_t = \prod_{j=1}^{t}\frac{T - j + 2}{T - j + 1}\alpha_0 = \frac{T + 1}{T - t + 1}.$$

We estimate $\frac{1}{\alpha_T}$ as

$$\frac{1}{\alpha_T} = \frac{T - T + 1}{T + 1} = \frac{1}{T + 1}.$$

We next estimate $\frac{1}{\alpha_T}\sum_{t=0}^{T}\alpha_t$ as

$$\frac{1}{\alpha_T}\sum_{t=0}^{T}\alpha_t = \frac{1}{T + 1}\sum_{t=0}^{T}\frac{T + 1}{T - t + 1} = \sum_{t=0}^{T}\frac{1}{T - t + 1} \leq 1 + \ln(T + 1).$$

Plugging these estimates back into (B.12), we obtain

$$\mathbb{E}[g(\mathbf{x}_{T+1}) - g^*] \leq \frac{1}{T + 1}\left(g(\mathbf{x}_0) - g^* + \frac{1}{2\tau}\|\mathbf{x}_0 - \mathbf{x}^*\|^2\right) + 2\tau L_g^2(1 + \ln(T + 1)).$$

Now, we use the definition of $\tau = \frac{1}{\sqrt{T+1}}$ to obtain the final bound for the last iterate:

$$\mathbb{E}[g(\mathbf{x}_{T+1}) - g^*] \leq \frac{g(\mathbf{x}_0) - g^*}{T + 1} + \frac{\|\mathbf{x}_0 - \mathbf{x}^*\|^2}{2\sqrt{T + 1}} + 2L_g^2\frac{1 + \ln(T + 1)}{\sqrt{T + 1}}.$$

We obtain the given bound by absorbing the constants. $\qquad\square$

## B.4. Proof for Section 4.4

*Proof of Theorem 4.4.* By using a constant step size $\tau$ in Lemma 5.3 of Lin et al. (2025) we have the inequality

$$\mathbb{E}_t\|\mathbf{x}_{t+1} - \mathbf{u}\|^2 \leq \|\mathbf{y}_t - \mathbf{u}\|^2 - 2\tau[g(\mathbf{x}_t) - g(\mathbf{u})] + 2mL_g\tau^2\|\nabla f(\mathbf{x}_t)\| + 2m^2L_g^2\tau^2$$

for all $\mathbf{u}$ measurable with respect to $\mathcal{F}_t$. Setting $\mathbf{u} = \mathbf{z}_t$ gives

$$\mathbb{E}_t\|\mathbf{x}_{t+1} - \mathbf{z}_t\|^2 \leq \|\mathbf{y}_t - \mathbf{z}_t\|^2 - 2\tau[g(\mathbf{x}_t) - g(\mathbf{z}_t)] + 2mL_g\tau^2\|\nabla f(\mathbf{x}_t)\| + 2m^2L_g^2\tau^2. \tag{B.13}$$

Using Young's inequality, we can estimate the third term on the right-hand side as

$$2mL_g\|\nabla f(\mathbf{x}_t)\| \leq \|\nabla f(\mathbf{x}_t)\|^2 + m^2L_g^2. \tag{B.14}$$

Using the definition of $\mathbf{y}_t = \mathbf{x}_t - \tau\nabla f(\mathbf{x}_t)$, we expand

$$\begin{aligned}\|\mathbf{y}_t - \mathbf{z}_t\|^2 &= \|\mathbf{x}_t - \mathbf{z}_t\|^2 - 2\tau\langle\nabla f(\mathbf{x}_t), \mathbf{x}_t - \mathbf{z}_t\rangle + \tau^2\|\nabla f(\mathbf{x}_t)\|^2 \\ &\leq \|\mathbf{x}_t - \mathbf{z}_t\|^2 - 2\tau(f(\mathbf{x}_t) - f(\mathbf{z}_t)) + \tau^2\|\nabla f(\mathbf{x}_t)\|^2, \tag{B.15}\end{aligned}$$

where in the last line we used the convexity of $f$. Combining (B.13), (B.14) and (B.15) yields

$$\mathbb{E}_t \|\mathbf{x}_{t+1} - \mathbf{z}_t\|^2 \leq \|\mathbf{x}_t - \mathbf{z}_t\|^2 - 2\tau[h(\mathbf{x}_t) - h(\mathbf{z}_t)] + 2\tau^2 \|\nabla f(\mathbf{x}_t)\|^2 + 3m^2 L_g^2 \tau^2.$$

By Lin et al. (2025, Lemma 5.7) or Lemma A.2, we obtain

$$\|\nabla f(\mathbf{x}_t)\|^2 \leq (1+\gamma)2L[h(\mathbf{x}_t) - h^*] + \left(1 + \frac{1}{\gamma}\right) \|\nabla f(\mathbf{x}^*)\|^2,$$

that is true for any $\gamma > 0$.

Hence, we have the inequality

$$\mathbb{E}_t \|\mathbf{x}_{t+1} - \mathbf{z}_t\|^2 \leq \|\mathbf{x}_t - \mathbf{z}_t\|^2 - 2\tau[h(\mathbf{x}_t) - h(\mathbf{z}_t)] + 4L(1+\gamma)\tau^2[h(\mathbf{x}_t) - h^*]$$
$$+ 2\left(1 + \frac{1}{\gamma}\right)\tau^2 \|\nabla f(\mathbf{x}^*)\|^2 + 3m^2 L_g^2 \tau^2. \tag{B.16}$$

Rearranging and dividing both sides of the above inequality by $2\tau$ gives

$$(1 - 2L(1+\gamma)\tau)\, h(\mathbf{x}_t) - h(\mathbf{z}_t) + 2L(1+\gamma)\tau\, h^*$$
$$\leq \frac{1}{2\tau}\|\mathbf{x}_t - \mathbf{z}_t\|^2 - \frac{1}{2\tau}\mathbb{E}_t\|\mathbf{x}_{t+1} - \mathbf{z}_t\|^2 + \left(1 + \frac{1}{\gamma}\right)\tau\|\nabla f(\mathbf{x}^*)\|^2 + 2m^2 L_g^2 \tau.$$

In the last term, we replaced $\frac{3}{2}$ with 2 since $m^2 L_g^2 \tau > 0$. Now, by the optimality condition for $h = f + g$, we have $-\nabla f(\mathbf{x}^*) \in \partial g(\mathbf{x}^*)$. Moreover, since each $g_j$ is $L_g$-Lipschitz, their sum $g$ is $mL_g$-Lipschitz, hence every subgradient of $g$ has norm at most $mL_g$. Thus, we have

$$\|\nabla f(\mathbf{x}^*)\| \leq mL_g.$$

Using this in Equation (B.16) gives

$$(1 - 2\tau L(1+\gamma))\, h(\mathbf{x}_t) - h(\mathbf{z}_t) + 2\tau L(1+\gamma)h^*$$
$$\leq \frac{1}{2\tau}\|\mathbf{x}_t - \mathbf{z}_t\|^2 - \frac{1}{2\tau}\mathbb{E}_t\|\mathbf{x}_{t+1} - \mathbf{z}_t\|^2 + \left(3 + \frac{1}{\gamma}\right)m^2 L_g^2 \tau.$$

By taking $\mu = 1 - 2\tau L(1+\gamma)$, $b = -1$, $c = 2\tau L(1+\gamma)$ and $v = \left(3 + \frac{1}{\gamma}\right)m^2 L_g^2 \tau$, we obtain

$$\mu h(\mathbf{x}_t) + bh(\mathbf{z}_t) + ch^* \leq \frac{1}{2\tau}\|\mathbf{x}_t - \mathbf{z}_t\|^2 - \frac{1}{2\tau}\mathbb{E}_t\|\mathbf{x}_{t+1} - \mathbf{z}_t\|^2 + v. \tag{B.17}$$

This expression has the recursive structure of Garrigos et al. (2025, Lemma 4.2) with different coefficients. To conclude, we verify the conditions of Garrigos et al. (2025, Lemma 4.3):

- $\mu + b + c = 0$: holds by construction.

- $\mu > 0$: holds when $\tau L < \frac{1}{2}$ (which is assumed) and by chosing $\gamma = (1 - 2\tau L)/(1 + 2\tau L) > 0$. This choice gives us:

$$\mu = \frac{1 - 2\tau L}{1 + 2\tau L}, \quad b = -1, \quad c = \frac{4\tau L}{1 + 2\tau L}, \quad v = \frac{4 - 4\tau L}{1 - 2\tau L}m^2 L_g^2 \tau < \frac{4}{1 - 2\tau L}m^2 L_g^2 \tau.$$

- $b \leq 0$: holds.

Additionally, notice that the term $v$ in our case is identical to the one in Garrigos et al. (2025, Lemma 4.2) up to the constant coefficient ($4m^2 L_g^2$ in ours vs $\sigma_*^2$ in theirs). Hence, we can apply Garrigos et al. (2025, Theorem 3.1) and Garrigos et al. (2025, Corollary 3.4) to obtain the $\mathcal{O}\left(\frac{\ln(T+1)}{\sqrt{T}}\right)$ rate. $\qquad\square$

# C. Additional experimental results

Below, we provide additional experimental results for the synthetic lasso problem using BlockProx. Additional details regarding experimental setup can be found in Section 5 in the main text.

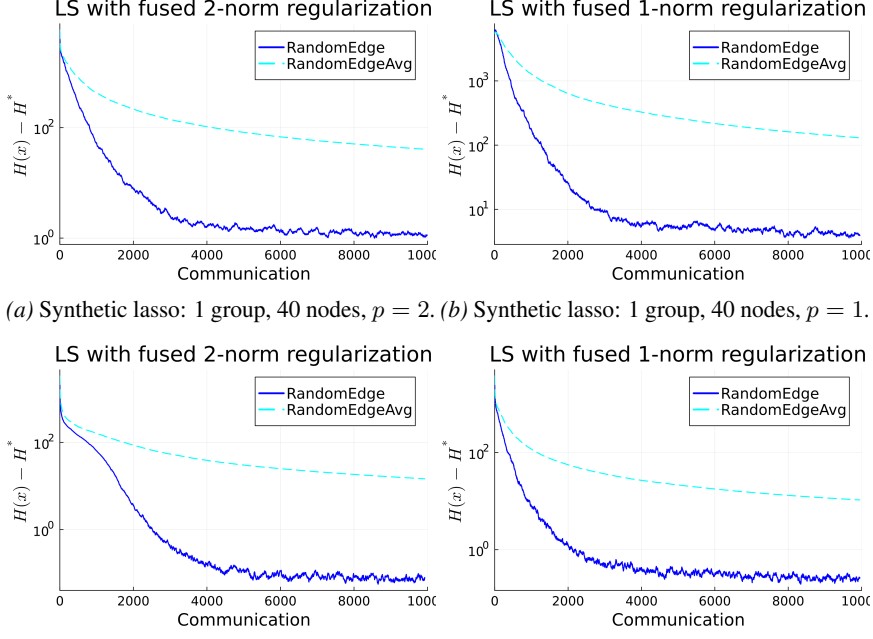

*(a)* Synthetic lasso: 1 group, 40 nodes, $p = 2$. *(b)* Synthetic lasso: 1 group, 40 nodes, $p = 1$.

*(c)* Synthetic lasso: 1 group, 20 nodes, $p = 2$. *(d)* Synthetic lasso: 1 group, 20 nodes, $p = 1$.

*Figure 3.* Additional results for the synthetic lasso problem using BlockProx.

