# OpenReview forum: "Convergence Rate of the Last Iterate of Stochastic Proximal Algorithms"
_ICML.cc/2026/Conference — ICML 2026 regular_

### Official Review · Reviewer_kHVG · 2026-03-11

**Soundness:** 3
**Presentation:** 3
**Significance:** 3
**Originality:** 3
**Overall Recommendation:** 5
**Confidence:** 4

**Summary:**

In this paper, the authors provide last-iterate convergence guarantees for Stochastic Proximal Gradient Descent, the Stochastic Proximal Point Method, and the Randomized Incremental Proximal Method. A primary contribution of this work is demonstrating that these methods converge with a near-optimal iteration complexity of  $\widetilde{\mathcal{O}}\big(\frac{1}{\sqrt{T}}\big)$ under component-wise convexity and smoothness, notably without relying on the restrictive bounded variance assumption.

**Compliance With Llm Reviewing Policy:**

Affirmed.

**Final Justification:**

Given the strength and rigor of its theoretical foundations, I recommend this paper for acceptance.

**Key Questions For Authors:**

No questions

**Limitations:**

No limits

**Strengths And Weaknesses:**

### **Strengths**

1. **Clarity and Motivation:** The paper is well-motivated and well-written. The authors clearly explain the theoretical challenges and the underlying motivation for their approach. Additionally, providing proof sketches directly in the main text significantly smooths the presentation and aids readability.

2. **Theoretical Contributions:** The submission presents strong and rigorous theoretical results.

### **Weaknesses**

**1. Literature Review:** While not a major flaw, the paper's contextualization would be significantly strengthened by mentioning the recent series of works focusing on Stochastic Proximal Point Methods [1-4]. Additionally, regarding the relaxation of standard assumptions, the authors should consider discussing [5], which also addresses optimization under individual smoothness without relying on bounded gradient assumptions.

**2. Minor Typo:** * On line 658, the term $2\tau$ appears to be missing.

**References:**
* [1] Khaled, A. and Jin, C., Faster federated optimization under second-order similarity. In *The Eleventh International Conference on Learning Representations*.
* [2] Traoré, C., Apidopoulos, V., Salzo, S. and Villa, S., 2024. Variance reduction techniques for stochastic proximal point algorithms. *Journal of Optimization Theory and Applications*, 203(2), pp.1910-1939.
* [3] Richtárik, P., Sadiev, A. and Demidovich, Y., 2024. A unified theory of stochastic proximal point methods without smoothness. *arXiv preprint arXiv:2405.15941*.
* [4] Sadiev, A., Condat, L. and Richtárik, P., 2024. Stochastic proximal point methods for monotone inclusions under expected similarity. *arXiv preprint arXiv:2405.14255*.
* [5] Lei, Y., Hu, T., Li, G. and Tang, K., 2019. Stochastic gradient descent for nonconvex learning without bounded gradient assumptions. *IEEE transactions on neural networks and learning systems*, 31(10), pp.4394-4400.

---

> ### Author Rebuttal · Authors · 2026-03-31
>
> Thank you for the constructive feedback! We hope that the clarifications provided below resolves your questions.
>
> # 1) On the literature review
>
> We will include discussions of these works in the related work section of the final version of the paper. Regarding [1]-[4], let us emphasize that none of these results can apply to our setting but we agree that including them in our literature survey will improve our paper. Regarding [5], this paper’s results are orthogonal to ours since they focus on unconstrained nonconvex optimization and do not get last iterate rates (or they use PL). We believe it is most related to the work of [Khaled and Richtarik, 2023] (from our paper’s bibliography) and we will discuss [5] along with this work.
>
> # 2) On the minor typo
>
> We thank the reviewer for pointing out this typo. It does not affect future steps in the proof. We will fix it in the final version of the paper.

---

> > ### Author Rebuttal · Reviewer_kHVG · 2026-04-01
> >
> > Thank you to the authors for their response. I appreciate the commitment to incorporating the suggested literature ([1-5]) into the related work section and for clarifying the distinctions between [5] and the current submission regarding the nonconvex setting and last-iterate rates.
> >
> > The authors also addressed the minor typo identified in the appendix. Since my initial concerns were primarily related to contextualization and presentation, and the authors have addressed these points satisfactorily, I am maintaining my original positive score. The theoretical contributions remain strong, and the proposed changes will improve the final quality of the paper.

---

> > > ### Author Response · Authors · 2026-04-04
> > >
> > > Thank you for the positive evaluation of our work and their acknowledgement. We appreciate the feedback and will incorporate the suggested literature into the related works section to further improve clarity. We will also make sure to fix the minor typo in the appendix.
> > >
> > > Thank you again for your helpful feedback.

---

### Official Review · Reviewer_9Zww · 2026-03-13

**Soundness:** 3
**Presentation:** 2
**Significance:** 2
**Originality:** 2
**Overall Recommendation:** 3
**Confidence:** 3

**Summary:**

This paper focuses on composite convex optimization where the objective is the sum of a smooth term and a nonsmooth regularizer and studies two algorithms: the proximal stochastic gradient method for a single regularizer and the randomized incremental proximal method when the regularizer is decomposable. The authors relax the bounded variance assumptioni and establish the $\tilde{O}(1/\sqrt{T})$ converegence rate for the last iterate.

**Compliance With Llm Reviewing Policy:**

Affirmed.

**Final Justification:**

The authors' rebuttal addresses my concerns. I believe the results of this paper have some value, and the proofs also contain some novelty, but I do not think these contributions are substantial. In addition, I think there is still room for improvement in the presentation.

**Key Questions For Authors:**

1. What are the similarities and differences between the proofs in Sections 3 and 4? Such a comparison would help readers understand the two methods.

2. What is the intuition for the auxiliary sequence $z_t$ and how is it specified?

3. Could the analysis in this paper be generalized to the decreasing step size? The number of iterations may not be known in advance.

**Limitations:**

Yes.

**Strengths And Weaknesses:**

[strengths]
1. Table 1 is very clear and helps clarify the improvement of this paper over previous work. In particular, this paper achieves the desired last-iterate convergence rate satisfying the followings simultaneouly (1) under relaxed variance condition, (2) dealing with possibly stochastic proximal operator.


[weaknesses]
1. There is some room for improvement in the presentation. For example
- It is a bit strange to discuss the detailed technical contribution in the section on Problem setting. Maybe it is better to be combined with the subsequent proof part.
- In Sections 3 and 4, the theoretical results and the proofs are mixed, bringing difficulty in understanding this paper. Maybe the authors could first present the main results and then discuss the significance. After that, summarize the proof sketch.

2. The authors mention graph-guided regularizers and network Lasso in the abstract and introduction, implying that this example is an important motivation. However, there are no corresponding numerical experiments for this setup. Moreover, it would be better to dicsuss why last-iterate convergence is important for such problems to support the motivation and significance.


3. The proof seems to be a direct generalization of previous work. Although the authors have discussed the main obstacles, it is not convincing enough.

---

> ### Author Rebuttal · Authors · 2026-03-31
>
> Thank you for the feedback. We hope that our clarifications resolves your questions.
>
> # 1) Regarding presentation
> >“combined with the subsequent proof”
>
> To improve clarity, we will move Sec 2.1 to the end of Sec 2 and change the title of Sec 2.
>
> >“Sections 3 and 4”
>
> Thank you for the suggestion! In fact, this is precisely the structure we adopted in Sec 3.1 and 3.2. In Sec 3.1, we introduce our main theoretical result and then discuss its significance. In Sec 3.2 we discuss proof sketch and technical ideas. Sec 4 mirrors the same structure.
>
> # 2) On graph-guided regularizers
> In our response to Reviewer hSom we present results of the BlockProx algorithm comparing the averaged iterate and the last iterate.
>
> > “Importance of last-iterate convergence”
>
> In many ML applications, the last iterate is the natural output. We believe it is folklore in many different areas of first-order optimization algorithms that last iterate works better. Indeed, this is why there is a long-lasting literature specifically for last iterate guarantees (See for example Shamir&Zhang’s work from 2013, Jain et al.’s work from 2019 and so on).
>
> Additionally, for regularizers that encourage sparsity such as the $\ell_1$ norm, the last iterate is often more desirable since it can yield a sparse solution, while the averaged iterate may not. We will further clarify these points.
>
> # 3) On novelty
> **We respectfully disagree with the reviewer that our work is a direct generalization of the previous work**. As the reviewer acknowledges, **we already talk about the main obstacles, showing that our work is not a direct generalization**. To further clarify the differences, we compare our work to some existing literature:
>
> - Compared to Garrigos et al., whose result focuses on SGD, we focus on the proximal setting. This extension is non-trivial since the prox operator is nonlinear which brings additional challenges in the analysis that is highlighted in Sec 2.1 of our paper. In SGD, we are able to utilize $\mathbb{E}[\nabla f_i(x^*)] = \nabla f(x^\star) = 0$. This is no longer true in our setting since $0 \in \nabla f(x^\star) + \partial g(x^\star)$. The added constraints also require additional mechanisms to handle which is not trivial.
>
> - Compared to Bertsekas, we avoid bounded variance and show last iterate rates. For example, in Lasso, the variance of the stochastic gradient $a_i(a_i^T x - b_i)$ can be unbounded, since $x$ is generally unbounded. In contrast, our result can handle such a setting and still guarantee the last-iterate convergence.
>
> - Lin et al., proves the last iterate for the BlockProx algorithm under a bounded gradient assumption for $f$. In contrast, we only assume smoothness of $f$ which is much weaker. Additionally, their experiments output the last iterate which performs much better than the averaged iterate (which we show in our response to Reviewer hSom).
>
> We emphasize that **our result is the first that can give guarantees on the most standard way of solving large scale instances of Lasso with proximal SGD that outputs the last iterate, since the previous bounded variance assumptions are not satisfied here. We believe that such a result is of *fundamental importance* that should not be dismissed.**
> On top of all these, we can even handle randomized prox operators, in a stochastic proximal-point type framework.
>
> # 4) Questions for the author
> 1)  Both proofs follow a similar structure using the auxiliary sequence $z_t$ to obtain the last iterate. However, the key difference is how we handle the stochasticity in $g$. **Sec 4.2 (until Lem 4.2) highlights how we use Lipschitzness of $g$ to handle this term which is the main difference.** This gives rise to additional terms that need to be handled and App B explains how we control them in further detail.
>
> 2) The auxiliary sequence $z_t$ is a standard tool used in the analysis of last iterate convergence, pioneered by Zamani&Glineur. Unfortunately, this is a technical idea and there is not much intuition behind it. We keep track of the distance between the reference point $z_t$ which is a convex combination of our iterates and $x_t$. The weights in the convex combination are chosen in a way that allows us to zero out all the previous objective gap terms and only keep the last gap term. This allows us to obtain the last iterate convergence rate. More details about the choice of the weights can be found in the proof of Thm 3.1 in App A.2.
>
> 3) Good point! We believe this extension is possible but is non-trivial and beyond the scope of the current paper. We we did not explore decreasing step size because the prior works on last iterate convergence (e.g., Zamani and Glineur, Garrigos et al. etc) use a constant step size. We wanted to first establish the last iterate rate for proximal algorithms under the same step size regime before exploring the more challenging decreasing step size case. We will mention this as an open question.
>
> Please let us know if you have further questions!

---

> > ### Author Rebuttal · Reviewer_9Zww · 2026-04-03
> >
> > Thanks for the detailed responses. I will raise my score to 3. However, I think there is still some room for improvement in the presentation. For example, since the proofs in Sections 3 and 4 are quite similar, it may be better to summarize them together while highlighting their main differences. This could leave more space in the main text for additional numerical results.

---

> > > ### Author Response · Authors · 2026-04-04
> > >
> > > Thank you for the helpful suggestion. We agree that section 3 and 4 share structural similarities. That said, we **deliberately chose to separate the two analyses** as they are focusing on **fundamentally different problem settings with distinct applications which we wished to emphasize.**
> > >
> > > In section 3, our focus was on proximal SGD where the proximal operator is not stochastic. The main purpose of this section was to highlight the deviation from the unconstrained SGD case (e.g. Garrigos et al) due to non linearity of the proximal operator. We also provide a clear outline of the proof and derive some useful corollaries such as for projected SGD. **Separating out this section allows us to clearly communicate our core technical ideas and how they extend recent last-iterate work to the proximal setting.**
> > >
> > > In section 4, we build on this foundation by adding stochasticity in the proximal operator. This leads to a fundamentally different setting with different applications. We made this section separate from the previous one to emphasize the applications in graph guided regularizer problems and for evaluating the BlockProx algorithm. Moreover, the analysis in this section requires **additional assumptions** (e.g. Lipschitzness) and techniques which do not arise in section 3. We therefore chose to present this section separately to **emphasize the new challenges and the several applications that are not covered in the setting of Section 3.**
> > >
> > > We felt that combining these two sections together could confuse readers regarding which results and applications apply to each setting. We also wanted to emphasize that our results in section 4 also extend to the BlockProx algorithm which is interesting in its own right. Additionally, the two sections rely on different assumptions and require different ways to handle terms which we wished to emphasize.
> > >
> > > That said, **we are happy to revise the presentation to further clarify the similarities and differences between the two sections.**
> > >
> > > From your response, we understand that the remaining concern is about presentation. **We would therefore like to better understand the updated score of three.** The definition of this score states *“A paper with clear merits, but also some weaknesses, which overall outweigh the merits.”* Based on your feedback, it seems that the remaining concern is limited to presentation (and a small portion of it) and **we respectfully feel that this does not outweigh the merits of our work.**
> > >
> > > Moreover, the final version allows for an additional page to incorporate revisions, so **there will be sufficient space in the main text for additional numerical results.**
> > >
> > > Given that we addressed most of your concerns, can you please reconsider your score taking into account the definition of each score?

---

### Official Review · Reviewer_hSom · 2026-03-13

**Soundness:** 3
**Presentation:** 3
**Significance:** 2
**Originality:** 2
**Overall Recommendation:** 3
**Confidence:** 3

**Summary:**

The authors consider an important area in stochastic convex optimization, namely the last-iterate convergence behavior of stochastic proximal algorithms for composite problems. The paper studies proximal SGD when the objective is composed of a smooth convex term and a nonsmooth convex regularizer, and also studies a randomized incremental proximal method when the regularizer is additively decomposable. The main contribution is to establish near-optimal last-iterate convergence rates of order $\tilde O(1/\sqrt{T})$ without relying on the classical uniformly bounded variance assumption, replacing it with componentwise convexity/smoothness and a second-moment bound only at the solution. The authors also extend the analysis to projected SGD, stochastic proximal point, and BlockProx-type settings, and provide experiments suggesting that the last iterate can outperform averaged iterates in practice.

**Compliance With Llm Reviewing Policy:**

Affirmed.

**Key Questions For Authors:**

See weakness.

**Limitations:**

yes.

**Strengths And Weaknesses:**

Strengths:
A key merit is that the results avoid assuming globally bounded variance of stochastic gradients, and instead only require convexity and smoothness of each component (f_i), plus a finite second moment at a solution point. This is a more realistic setup for large-scale problems such as least-squares/Lasso-type models.

Weakness:
1. The RIPM result requires each $g_j$ to be Lipschitz, which restricts the applicability of the result.
2. The final rate is only improved up to logarithmic factors and is given in expectation rather than with high probability.
3. The empirical section includes only two illustrative tasks and lacks broader baselines.

---

> ### Author Rebuttal · Authors · 2026-03-31
>
> We thank the reviewer for their constructive feedback. We hope that the clarifications provided below would resolve your questions.
>
> # **1) On the Lipschitz assumption**
>
> We agree that the Lipschitz assumption in the RIPM result restricts applicability. However, we would like to emphasize that our main proximal SGD result in section 3 does not require this assumption and establishes a $\widetilde O(1/\sqrt{T})$ last iterate guarantee under the relaxed variance assumptions which is fundamental to problems such as Lasso.
>
> We believe that this result is already a significant contribution since it is the first result that can give guarantees on the most standard way of solving the fundamental problem of Lasso in the large-scale regime, with proximal SGD that outputs the last iterate, since the previous bounded variance assumptions are not satisfied here.
>
> Additionally, we would like to point out that the Lipschitz assumption is standard in prior work on stochastic proximal-point and incremental proximal methods (e.g., Bertsekas, 2011; Cai & Diakonikolas, 2025; Lin et al. 2025), and is not specific to our analysis.
>
> Moreover, this assumption is satisfied in several important applications, including norm-based regularizers and graph-guided regularization (as in Lin et al. 2025), which is one of our motivating examples. We also note that some related works impose even stronger assumptions such as smoothness/strong convexity (e.g., Traoré et al., 2024), which excludes common nonsmooth regularizers like the $\ell_1$ norm.
>
> # **2) On the convergence rate**
>
> We would like to clarify that our goal is not to improve the convergence rate beyond existing literature. Instead, our main contribution is extending the last iterate guarantees to the proximal setting under minimal assumptions which, to our knowledge, has not been addressed in the literature before.
>
> As far as we are aware, it is not known whether the best rate for the last iterate can be improved beyond $O(\frac{\log(T)}{\sqrt{T}})$ in the convex smooth setting. The closest work is Jain et al., 2019 which shows that the last iterate of SGD can achieve $O(\frac{1}{\sqrt{T}})$ under Lipschitzness and bounded variance assumptions. However, it is not clear whether this can be extended to the proximal setting under relaxed variance assumptions and without Lipschitzness.
>
> Let us remark that expectation guarantees are standard in the literature on stochastic optimization. Obtaining high probability guarantees typically require stronger assumptions than those used in our analysis and would require different techniques. We will consider this extension in future work. In particular, we believe that showing the in-expectation rates is the required first step before showing stronger results such as high probability rates. Our result improves the state-of-the-art in the literature for in-expectation rates.
>
> # **3) On the empirical section**
>
> We agree that additional experiments would further strengthen the paper. Our primary focus is theoretically establishing last-iterate convergence guarantees under relaxed assumptions which, to our knowledge, has not been addressed in the proximal setting.
> That said, we emphasize that the last iterate is the standard output in many machine learning and deep learning applications where model parameters are not averaged in practice. This makes last iterate guarantees fundamental.
>
> To better address this point, we will expand the empirical section by including additional experiments in the graph structured settings (BlockProx), where our theory applies. We will report comparisons between the last iterate and the averaged iterate to further support the motivation of our work. For more details about the experimental setup, please refer to [https://arxiv.org/pdf/2509.14488v1] section 6.1 and 6.2. We will also include these results in the final version of the paper. We include the anonymous links for the plots below:
>
> **Network of 5 groups - $\ell_1$ regularized**
> https://github.com/anon8345/last_iterate_prox_experiments/blob/main/Figures/LS_1_5_75.png
>
> **Network of 5 groups - $\ell_2$ regularized**
> https://github.com/anon8345/last_iterate_prox_experiments/blob/main/Figures/LS_2_5_75.png
>
> **Network of 1 group with 20 nodes - $\ell_1$ regularized**
> https://github.com/anon8345/last_iterate_prox_experiments/blob/main/Figures/LS_1_1_20.png
>
> **Network of 1 group with 20 nodes - $\ell_2$ regularized**
> https://github.com/anon8345/last_iterate_prox_experiments/blob/main/Figures/LS_2_1_20.png
>
> **Network of 1 group with 40 nodes fully connected - $\ell_1$ regularized**
> https://github.com/anon8345/last_iterate_prox_experiments/blob/main/Figures/LS_1_1_40.png
>
> **Network of 1 group with 40 nodes fully connected - $\ell_2$ regularized**
> https://github.com/anon8345/last_iterate_prox_experiments/blob/main/Figures/LS_2_1_40.png
>
> **Housing dataset Results**
> https://github.com/anon8345/last_iterate_prox_experiments/blob/main/Figures/housing.png

---

### Official Review · Reviewer_idH4 · 2026-03-15

**Soundness:** 3
**Presentation:** 3
**Significance:** 3
**Originality:** 2
**Overall Recommendation:** 4
**Confidence:** 5

**Summary:**

The paper extends recent last-iterate convergence results beyond unconstrained SGD to stochastic proximal methods for convex composite optimization. It proves an $1/\sqrt{T}$ (ignoring log factors) last-iterate rate for proximal SGD without assuming uniformly bounded gradient variance, requiring only smoothness and a bounded second moment at the optimum. The analysis is also extended to projected SGD and randomized incremental proximal methods, covering structured regularized problems.

**Compliance With Llm Reviewing Policy:**

Affirmed.

**Final Justification:**

In light of the author's rebuttal, I maintain my positive evaluation of 4.

**Key Questions For Authors:**

See Limitations.

**Strengths And Weaknesses:**

**Strengths**
- Proximal points methods are important for solving regularized ERM problems in ML, and relevant in the boarder setting of modern optimization, therefor the problem analyzed in this paper is significant.

- The paper gives a clean extension of recent last-iterate analyses from unconstrained SGD to the proximal / composite convex setting. It avoids the standard uniformly bounded variance assumption, replacing it with a weaker requirement based on the second moment at the optimum. The obtained rates are nearly optimal in the iteration counter.

- Authors makes a good work in explaining where the analysis from Zamani & Glineur (and later development from Garrigos et al. and Attia et al.) need to be adapted.

**Weaknesses**
- One concern is that extending the last-iterate analysis to proximal gradient methods under the minimal variance assumption does not seem to require a substantial departure from the proof techniques of Zamani and Glineur (with which I am familiar). This is not necessarily a flaw, but it positions the work primarily as a technical extension and does not seem to add much to our current understanding of proximal algorithms.

- I found the discussion around $N$-independence somewhat unconvincing. While the proposed bounds are indeed agnostic to dataset size, it is unclear why this should be viewed as a decisive advantage for regularized ERM, where the finite-sum structure is central. In particular, bounds that depend on $N$ are not inherently weaker; such dependence may simply reflect that the analysis leverages finite-data structure, which can yield genuinely sharper behavior in multi-pass or without-replacement regimes. As written, the paper does not sufficiently clarify in which practical or theoretical regimes $N$-independence should be preferred over more structured finite-sum guarantees.

- Theorem 4.1, which deals with an interesting class of problems, features a rather problematic dependence on $m^2$. This seems particularly critical in settings such as the graph-based regularizers used as motivating examples in the introduction: if $m$ denotes the number of edges, then $m$ may already be of the order of the square of the number of learning tasks. In that case, the bound in Theorem 4.1 could easily become vacuous unless $T$ is extremely large.

- I am a bit surprised that the experimental evaluation does not include graph-based regularizers, since these are presented as motivating examples in the introduction.

---

> ### Author Rebuttal · Authors · 2026-03-31
>
> Thank you for the constructive feedback! We hope that the clarifications provided below resolves your questions. Please let us know if you have any other questions!
>
> # 1) About Zamani and Glineur [ZM25]
>
> We agree that our analysis builds on the framework of [ZM25]. We respectfully disagree that our work does not add much to our understanding of proximal algorithms. We believe that **showing that the analysis of proximal algorithms without the bounded variance assumption can incorporate the idea of [ZM25], is indeed adding to our understanding of proximal algorithms**. In particular, we use the idea of [ZM25] just as many works in the literature used this idea, such as Cai&Diakonikolas, Liu&Zhou, Garrigos et al., Attia et al and so on. We believe all these works add to our knowledge of either SGD or proximal methods. We will be careful to position our paper by explaining technical contributions.
>
> The main tool in [ZM25] is the use of the auxiliary sequence $z_t$ which allows them to zero out all the previous objective gap terms and only keep the last objective gap term to obtain the last iterate rate.
>
> While we use the same tool in our analysis, **their setting and assumptions are quite different from ours**, and extending their techniques to the **proximal setting under relaxed variance assumptions requires addressing several non-trivial challenges**. Their analysis considers projected subgradient methods in the deterministic setting, when subgradients are uniformly bounded. In contrast, we consider the stochastic proximal setting where the variance can be unbounded, and we assume only smoothness of the objective function. This setting is fundamentally different and requires different techniques to handle the additional terms that arise in the analysis.
>
> While we build on the idea of tracking the distance between the iterates and an auxiliary sequence, the way we build our recurrence relation is quite different from theirs. Moreover, we handle additional complications such as incorporating stochastic proximal-point updates which are different from [ZM25]. Admittedly, for incorporating stochastic proximal-point type updates, we utilize ideas from that literature, but we believe that this is standard – one needs to use ideas from the literatures that the new developments build on.
>
> Overall, while the tool of using an auxiliary sequence and building a combination of objective gap terms is inspired by [ZM25], the way we arrive at this combination and handle the additional terms that arise require non-trivial adaptations of their techniques. Hence, we believe that our analysis provides new insights into the proximal setting and is not a straightforward extension of their work.
>
> # 2) $N$-independence
>
> We agree with the reviewer that $N$-independence is not inherently better than $N$-dependence. Our intention was not to claim that $N$-independence is always better, but rather to clarify that it can be advantageous in certain regimes, such as when the number of component functions is very large, or when only a single pass over the data is feasible. In such settings, bounds that scale with $N$ are less helpful, and $N$-independent bounds can provide more insight into the algorithm behaviour.
>
> For instance, Liu \& Zhou (2024) obtain the rate $\widetilde O(\frac{1}{n^{1/3}K^{2/3}})$ for the random reshuffle scheme where $n$ is the number of component functions and $K = T/n$ is the number of epochs. One obtains that our bound is better when $n > \sqrt{T}$ and since $T = nK$, our bound is better when the dataset size is larger than the number of epochs. This highlights a regime where $N$-independence can be advantageous.
>
> We also emphasize that we additionally allow randomized prox operators which is not allowed in the current results for shuffling methods.
>
> # 3) On the $m^2$ dependence in Thm 4.1
>
> This arises because we decomposed our regularizer as $g(x) = \sum_{i=1}^m g_i(x)$ which leads to the update $x_{t+1} = prox_{\tau m g_{j_t}}(\cdot)$ introducing an extra $m$ factor. However, if instead we consider the average formulation $g(x) = \frac{1}{m} \sum_{i=1}^m g_i(x)$, the update becomes  $x_{t+1} = prox_{\tau g_{j_t}}(\cdot)$ which removes the $m$ factor (similar to Cai&Diakonikolas). The reason we chose the former formulation is that it is more standard in the literature (e.g., Lin et al., Bertsekas). Both these works have the same dependence on $m$. Lin et al. who uses a similar iteration scheme as us has a $m^2$ dependence in the smooth regime (Theorem 5.9). Bertsekas et al. when we match his problem ($\sum (f_i + g_i)$) to our template also has the $m^2$ dependence.
>
> # 4) On the experimental evaluation
>
> In our response to Reviewer hSom, we present results of the BlockProx algorithm comparing the averaged and last iterate. For more details about the experiment, please refer [https://arxiv.org/pdf/2509.14488v1] section 6.1 and 6.2. We will also include these results in the final version of the paper.

---

> > ### Author Rebuttal · Reviewer_idH4 · 2026-04-01
> >
> > I thank the authors for their rebuttal and for addressing my questions. I also appreciate their effort in running additional experiments. However, more details would have been helpful, especially since the provided link does not appear to work.
> >
> > Finally, I would like to clarify one point: I never stated that the analysis is a straightforward extension of [ZM25]. Rather, in my review I noted that, from the current presentation, it is not entirely clear to me what the main novel insights about proximal algorithms are that this submission provides.
> >
> > That said, analyzing existing methods under more general assumptions, even when building on prior work, is valuable in its own right. In my opinion, this is precisely the contribution of the submission. My overall evaluation therefore remains positive.

---

> > > ### Author Response · Authors · 2026-04-04
> > >
> > > Thank you for your acknowledgement and maintaining your positive evaluation! We apologize for the typo in the link. The right bracket got accidentally included in the link which caused it to not work. The correct link is https://arxiv.org/pdf/2509.14488v1 . We will also include the additional experiments we conducted into our paper.
> > >
> > > Thank you also for your clarification! We will revise our introduction to better clarify our contributions and insights about proximal algorithms.

---

### Decision · Program_Chairs · 2026-04-30

**Decision:**

Accept (regular)

**Comment:**

This paper studies the last iterate convergence of stochastic proximal algorithm. In particular, the authors show a $O(log(T)/\sqrt{T})$ convergence rate for proximal SGD and randomised incremental proximal method.

The reviewers acknowledged the quality and the novelty of the contributions

The main concerns were regarding:
- the presentation: this was non-consensual among the reviewers, and I believe the presentation is sufficiently good for an ICML paper.
- the novelty: this was addressed.
- the experiments: this remains probably the weakest point of the paper. Given that the motivating applications are "graph-guided regularizers that arise in multi-task and federated learning", it would be useful to show that the proposed rate (i.e. last iterate convergence) are reasonnable for large-scale tasks. Since, the studied algorithms are pretty common, I do not see the lack of additional empirical validation as a sufficient reason for rejecting this paper but I would recommend the authors to add such experiments (e.g. last iterate convergence for graph-guided regularizers and network Lasso) in the revision of the paper.

Overall I believe the main concerns have been properly addressed by the authors, and I thus recommend this paper for acceptance.